



# Measurement report: High contribution of $N_2O_5$ uptake to particulate nitrate formation in $NO_2$-limited urban areas

Ziyi Lin[1,2,3], Chuanyou Ying[4], Lingling Xu[1,2*], Xiaoting Ji[1,2,3], Keran Zhang[1,2], Feng Zhang[2], Gaojie Chen[1,2,3], Lingjun Li[1,2,3], Chen Yang[1,2,3], Yuping Chen[1,2,3], Ziying Chen[1,2,3], Jinsheng Chen[1,2*]

**Affiliations:**

[1]State Key Laboratory of Advanced Environmental Technology, Institute of Urban Environment, Chinese Academy of Sciences, Xiamen 361021, China

[2]Fujian Key Laboratory of Atmospheric Ozone Pollution Prevention, Institute of Urban Environment, Chinese Academy of Sciences, Xiamen 361021, China

[3]University of Chinese Academy of Sciences, Beijing 100049, China

[4]Fuzhou Institute of Environmental Science, Fuzhou 350013, China

*Correspondence to: Jinsheng Chen (jschen@iue.ac.cn); Lingling Xu (linglingxu@iue.ac.cn)

**Abstract:** Particulate nitrate ($pNO_3^-$) is a major component of fine particle in Chinese urban areas. However, the relative contributions of $pNO_3^-$ formation pathways in $NO_2$-limited urban areas remain poorly quantified, hindering further particulate pollution control. In this study, comprehensive winter field observations were conducted in urban Xiamen, Southeast China. We observed significantly elevated nighttime $pNO_3^-$ levels concurrent with increased $N_2O_5$ concentrations. Quantification using an observation-constrained model revealed that $N_2O_5$ uptake contributed 51.2% to total $pNO_3^-$ formation, which was comparable to that of the $OH + NO_2$ reaction. The $N_2O_5$ uptake was found to be mainly driven by nocturnal $NO_3$ oxidation capacity (modulated by $NO_2$ and $O_3$ levels) rather than by heterogeneous reaction conditions. Sensitivity simulations further demonstrated that $pNO_3^-$ formation rate was more sensitive to $NOx$ variations than to VOCs variations. Implementing $NOx$ control measures at nighttime was shown to effectively reduce $pNO_3^-$ by abating $N_2O_5$ uptake while simultaneously preventing daytime $O_3$ increase. Our findings enhance the understanding of $pNO_3^-$ formation in $NO_2$-limited urban areas and provide valuable insights for developing joint $PM_{2.5}$ and $O_3$ mitigation strategies.



## 1 Introduction

Fine particulate matter ($PM_{2.5}$) contributes to various atmospheric environmental issues, including visibility deterioration, radiative forcing change, and adverse impacts on human health (Seinfeld, 1989; Lelieveld et al., 2015). Among its chemical components, particulate nitrate ($pNO_3^-$) has attracted increasing attention due to its rising mass fraction in $PM_{2.5}$ and its nonlinear responses to emission mitigation strategies (Xie et al., 2022; Zhai et al., 2021; Li et al., 2021; Zhang et al., 2021; Zhou et al., 2022; Zong et al., 2022; Wang et al., 2020). The primary formation pathways of $pNO_3^-$ include gas-phase oxidation through the reaction of hydroxyl radicals (OH) and nitrogen dioxides ($NO_2$) (R1–R2), and heterogeneous uptake of dinitrogen pentoxide ($N_2O_5$) which is produced via $NO_2$ oxidation by nitrate radicals ($NO_3$) (R3–R5) (Brown and Stutz, 2012). It is well recognized that the OH + $NO_2$ reaction dominates in daytime, while $N_2O_5$ uptake dominates in nighttime. During nocturnal $pNO_3^-$ formation, particulate chlorides can induce $N_2O_5$ heterogeneous uptake to produce $ClNO_2$, thereby competing with $pNO_3^-$ formation.

$$OH\ (g) + NO_2\ (g) + M \rightarrow HNO_3(g) + M \tag{R1}$$

$$HNO_3(g) + NH_3(g) \rightleftharpoons NH_4NO_3(p) \tag{R2}$$

$$NO_2(g) + O_3(g) \rightarrow NO_3(g) \tag{R3}$$

$$NO_2\ (g) + NO_3(g) \rightleftharpoons N_2O_5(g) \tag{R4}$$

$$N_2O_5(g) + H_2O/Cl^-(p) \rightarrow (2-\varphi)NO_3^-(p) + \varphi ClNO_2(g) \tag{R5}$$

Many studies have focused on quantifying the potential formation pathways of $pNO_3^-$ in urban areas of China. In major urban agglomerations such as the Beijing-Tianjin-Hebei (BTH) region (Chen et al., 2020; Ma et al., 2023; Zhao et al., 2023), Yangtze River Delta (YRD) (Sun et al., 2022; Zhai et al., 2023; Zhang et al., 2023b), and Pearl River Delta (PRD) (Yang et al., 2022; Niu et al., 2022; Cheng et al., 2024), $pNO_3^-$ formation was typically dominated by the gas-phase oxidation of OH + $NO_2$. In contrast, under special conditions such as the COVID-19 pandemic and $PM_{2.5}$ pollution events (Yan et al., 2023; Zhai et al., 2023), $N_2O_5$ uptake became the main pathway. Previous research has demonstrated that the formation rate of $pNO_3^-$ via $N_2O_5$ uptake is closely related to its precursor $NO_2$ and $O_3$, and the $N_2O_5$ formation can be classified into $NO_2$-limited and $O_3$-limited regimes based on the $NO_2/O_3$ ratio (Ma et al., 2023). The winter $NO_2/O_3$ ratios in the BTH, YRD, and PRD regions were generally above 1, placing $N_2O_5$ formation



in the $O_3$-limited or transition regime (Ma et al., 2023; Wen et al., 2018; Li et al., 2021; Zhang et al.,
2023b). However, $N_2O_5$ uptake served as the dominant pathway for $pNO_3^-$ formation, typically occurring
under $NO_2$-limited conditions (e.g., reduced emissions during the pandemic) or highly favorable $N_2O_5$
uptake conditions (e.g., severe particulate pollution episodes). Collectively, these findings indicate that
spatial variations in $NO_2$ and $O_3$ levels are likely a key driver of regional differences in the dominant
formation pathways of $pNO_3^-$. The formation of $pNO_3^-$ primarily depends on precursors OH, $NO_2$, and
$O_3$, with OH and $O_3$ concentrations being influenced by VOCs and NO$x$ emissions. Thus, the different
formation pathways of $pNO_3^-$ result in complex responses to NO$x$/VOCs emissions. As for the response
of OH + $NO_2$ to precursors variation, it was relatively well-understood, as most Chinese urban areas are
located in VOC-limited regimes for $O_3$ (Wang et al., 2023b; Wang et al., 2022c; Zhang et al., 2023a; Mao
et al., 2022), and ammonia-rich regimes for $pNO_3^-$ (Xing et al., 2018; Sun et al., 2022; Fu et al., 2024;
Liu et al., 2019). Under these conditions, VOCs reduction suppresses $pNO_3^-$ formation by decreasing OH
concentrations, whereas NO$x$ reduction enhances $pNO_3^-$ formation by weakening the NO$x$ titration effect.
Given the regional variations in the $NO_2/O_3$ ratio across urban areas of China (Ma et al., 2023), the
response of $N_2O_5$ uptake to precursor changes (VOCs, $O_3$) likely exhibits spatial heterogeneity. A recent
study has revealed that under $O_3$-limited conditions for $N_2O_5$ formation (Zhang et al., 2023b), NO$x$
emissions had negligible effects, while VOCs reduction decreased the removal of $NO_3$ by VOCs, thereby
enhancing $N_2O_5$ uptake. However, the response of $pNO_3^-$ formation to precursors under $NO_2$-limited
conditions remains unclear. Aside from precursor availability, $N_2O_5$ uptake is also greatly influenced by
heterogeneous reaction conditions like aerosol composition and aerosol surface area (Mcduffie et al.,
2018b; Mcduffie et al., 2018a; Tham et al., 2018; Yu et al., 2020), which introduces additional uncertainty
in determining the contribution of $pNO_3^-$ formation pathways and the effectiveness of precursor control
strategies.

85        The $NO_2/O_3$ ratios in southeastern China predominantly fell within the $NO_2$-limited regime for $N_2O_5$

formation (Ma et al., 2023). Xiamen, as one of the most developed cities in southeastern China, exhibits
relatively better air quality with low levels of VOCs and NO$x$ compared to China's megacities (**Table**
**S1**). This pattern well represents the future urban atmospheric conditions following the implementation
of air pollution control measures in China. From December 2022 to February 2023, we conducted
comprehensive multi-parameter observations in urban Xiamen, including $N_2O_5$ and related chemical
constituents. An observation-constrained box model incorporating the heterogeneous reaction parameters



was utilized to quantify the rates of different $pNO_3^-$ formation pathways. Explainable machine learning
(ML) method was applied to identify the driving factors of high $N_2O_5$ uptake rate. Additionally, multi-
scenario simulations were performed to examine the joint responses of $pNO_3^-$ and $O_3$ formation to various
NO*x* and VOCs emissions. These findings enhance our understanding of $pNO_3^-$ formation pathways and
their environmental implications in $NO_2$-limited regions, providing valuable insights for developing joint
$PM_{2.5}$ and $O_3$ mitigation strategies.

**2 Methods**
**2.1 Field Observation.**
Field observations were conducted during the winter period from 1 December 2022 to 3 February 2023,
at an urban site (marked by the red star in **Figure S1**) in Xiamen, which is located in the southeastern
coastal region of China. Detailed site information has been described in our previous studies (Yang et al.,
2023; Liu et al., 2022). Trace gases (including PAN, HCHO, HONO, VOCs, $O_3$, NO*x*, CO, and $SO_2$),
chemical components in $PM_{2.5}$ (including organic carbon and elemental carbon, $SO_4^{2-}$, $NO_3^-$, $NH_4^+$, $Cl^-$),
$PM_{2.5}$ mass concentration, and meteorological parameters (including ambient temperature (T), relative
humidity (RH), atmospheric pressure (P), wind speed (WS), wind direction (WD), and photolysis rates)
were continuously measured during the campaign. Detailed information about measurement methods and
instruments is summarized in **Text S1**. A chemical ionization time-of-flight mass spectrometer equipped
with an iodide source (iodide-TOF-CIMS, Aerodyne Research Inc., USA) was deployed to measure $N_2O_5$
and $ClNO_2$. The instrument configuration and calibration procedures for $N_2O_5$ and $ClNO_2$ are described
in **Text S2**, following established methods (Wang et al., 2022b; Wang et al., 2022a; Thaler et al., 2011).
Boundary layer height (BLH) data were obtained from the ERA5 dataset (Hersbach et al., 2020).

**2.2 Determination of $pNO_3^-$ Formation Rate.**
The interactive box model developed by Wagner et al. with a simplified mechanism was employed to
obtain key parameters of the $N_2O_5$ uptake process (Wagner et al., 2013), including $kN_2O_5$ and $\varphi ClNO_2$
(**see in Text S3**). To validate the interactive box model results, these parameters were calculated
concurrently based on the classical steady-state approximation method (**Text S4**) (Brown et al., 2003;
Chen et al., 2022). As shown in **Figure S2**, the outcomes of the two methods exhibited strong consistency,
with logarithmic correlation coefficients ($R^2$) as high as 0.76 and 0.73 for $kN_2O_5$, $\varphi ClNO_2$, respectively.



Considering the larger number of valid data points, the model-derived parameters were adopted for
subsequent analysis.

124       A Framework for 0-D Atmospheric Modeling (F0AM), incorporating the Master Chemical

Mechanism (MCM v3.3.1) and heterogeneous mechanisms (**Table S2**), was employed to simulate nitrate
formation rates for each day during the study period (Wolfe et al., 2016; Atkinson and Arey, 2003; Jenkin
et al., 2015). The heterogeneous parameters derived from the interactive box model were implemented
in F0AM. In addition, hourly interval data of trace gases, photochemically active species, meteorological
variables, and reanalysis data were also applied to constrain the multiphase chemical box model. Detailed
model configurations are provided in **Text S5**. As shown in **Figure S3**, the model performed well for
$N_2O_5$ and $ClNO_2$ simulations with $R^2$ of 0.88 and 0.49, respectively. The simulated OH concentrations
agreed well with parameterized method suggested by Ehhalt and Rohrer (**Figure S4**, $R^2 = 0.86$) (Ehhalt
and Rohrer, 2000). Based on model simulation and precursor observations, we quantified $pNO_3^-$
formation rates through both $OH + NO_2$ and $N_2O_5$ uptake pathways by model integral. Note that the gas-
particle partitioning coefficient was set to 100%, which might lead to in an overestimation of the $OH +$
$NO_2$ pathway contribution.

**2.3 Identification of influencing factors for $N_2O_5$ uptake.**
Extreme gradient boosting (XGBoost), a machine learning technique, has been widely applied in
atmospheric chemistry research (Gui et al., 2020; Wang et al., 2023c; Requia et al., 2020). Here, we built
a XGBoost model to reproduce the $N_2O_5$ uptake rate with selected variables. The model was built using
the "xgboost" library (https://github.com/dmlc/xgboost/tree/master) in a python environment.
Explanatory variables included meteorological parameters (BLH, T, and RH), nocturnal atmospheric
oxidation capacity $P(NO_3)$ calculated by $k_{NO_2+O_3}[NO_2][O_3]$, TVOCs, the logarithm of the ratio of $NO_2$
to $O_3$ ($log([NO_2]/[O_3])$), NO, and heterogeneous uptake parameters ($\varphi ClNO_2$ and $kN_2O_5$). Only nighttime
(18:00 – 06:00 the next day) data were considered to identify key drivers of $N_2O_5$ uptake. The
hyperparameters of the XGBoost model were tuned by grid searching method and the established model
was evaluated using $R^2$, Mean Absolute Error (MAE) and Root Mean Square Error (RMSE). By
incorporating SHAP interpretation, the XGBoost-SHAP method could quantify factor contributions
through SHAP values, where absolute SHAP values denote the relative importance. Detailed description
and setup of the XGBoost-SHAP method can be found in **Text S6** and our previous study (Lin et al.,



152     2024).


**2.4 Emission Scenario Modelling.**
Using the aforementioned multiphase chemical box model, we investigated changes in formation rates
of $pNO_3^-$ ($PNO_3^-$) and $O_3$ ($PO_3$) under different VOCs and NO$x$ emission scenarios. The base model
simulation was performed using mean diurnal values from the winter 2022 observations. A series of
emission scenarios were tested by scaling normalized VOCs and NO$x$ concentrations from 0 to 2 times
baseline levels to examine their impacts on $PNO_3^-$ and $PO_3$. Prior to each scenario simulation, 3-day spin-
up was set to stabilize intermediate species concentrations. Isopleth diagrams of simulated $PNO_3^-$ and
$PO_3$ were obtained from the base scenario and 120 emission change scenarios. In addition, response
strength (RS) was calculated using **eq 2** as an indicator of emission sensitivity.
$$PO_3 = k_1[HO_2][NO] + \sum k_{2i}[RO_2][NO] \tag{1}$$
Where, $k_i$ is the corresponding chemical reaction rate constants.
$$RS = \frac{X_i - X_{base}}{V_i - V_{base}} \tag{2}$$
Where, $X_i$ and $X_{base}$ are the mean formation rates of dependent variables e.g. $PNO_3^-$, $PO_3$ in scenario i
and base simulations, respectively. $V_i$ and $V_{base}$ are the emission rates for the scenario i and base
simulations, respectively. Notably, the emission rates ranged from 0 to 2 times baseline levels, with the
base simulation emission rate normalized to 1.

**3 Results and Discussion**
**3.1 Overview of Observations.**
The mean diurnal patterns of $pNO_3^-$, gaseous pollutants and relevant meteorological parameters are
shown in **Figure 1**. During the entire observation period, mean concentrations of $NO_2$, $O_3$, total VOCs,
and $PM_{2.5}$ were 10.9 ppb, 27.3 ppb, 18.2 ppb, and 14.3 μg m$^{-3}$, respectively, lower than those observed
in most of China's key cities (refer to **Table S1**). Despite the low NO$x$ levels, $pNO_3^-$ contributed 29.5%
to $PM_{2.5}$ mass concentration, which was higher than proportions reported in Beijing urban area (24.7%)
(Ma et al., 2023), Guangdong (24.0%) (Yun et al., 2018), and Nanjing (24%–27%) (Huang et al., 2020).
This discrepancy suggests efficient conversion from $NO_2$ to $pNO_3^-$ in the study area. In addition, the
proportion of $pNO_3^-$ increased with rising $PM_{2.5}$ concentration (**Figure S6**), indicating its importance to



particulate pollution. This is consistent with the phenomenon widespread in urban areas of China where
$pNO_3^-$ became dominant in inorganic aerosols despite $NOx$ reduction, underscoring the need for efficient
$pNO_3^-$ control strategies (Zhai et al., 2021; Zhao et al., 2020; Zhang et al., 2022).
The diurnal pattern of $pNO_3^-$ exhibited a bimodal characteristic, with peaks occurring at 4:00 and
16:00 LT, respectively. The daytime peak (07:00–17:00) was accompanied by low concentrations of $NOx$
and high levels of $O_3$ and $JNO_2$, indicating that active photochemical conditions promoted daytime $pNO_3^-$
formation. During the nighttime (18:00–06:00 the next day), $pNO_3^-$ concentrations increased together
with $NO_2$, $N_2O_5$ and $ClNO_2$ from 18:00 onward and remained elevated until early morning. This
nighttime accumulation can be attributed to two factors. First, lower temperature, shallower boundary
layer height, and reduced wind speed at night favored the accumulation of $pNO_3^-$ and related nitrogen-
containing species. Second, higher RH and $PM_{2.5}$ concentrations at night enhanced aerosol water content
and surface area, providing favorable conditions for heterogeneous hydrolysis of $N_2O_5$ to form $pNO_3^-$.
The mean concentration of $N_2O_5$ was $0.19 \pm 0.26$ ppb (peaking at 2.52 ppb), which is relatively higher
than values reported for China's megacities (Chen et al., 2020; Wang et al., 2017; Tham et al., 2018;
Wang et al., 2022a; Liu et al., 2025; Li et al., 2023). Moreover, the observed elevation in nighttime $ClNO_2$,
primarily produce via the reaction of $N_2O_5$ with Cl-containing particles, strongly supports the presence
of active heterogeneous processes of $N_2O_5$. Collectively, these findings imply a likely significant
contribution of $N_2O_5$ uptake to $pNO_3^-$ formation during the nighttime.

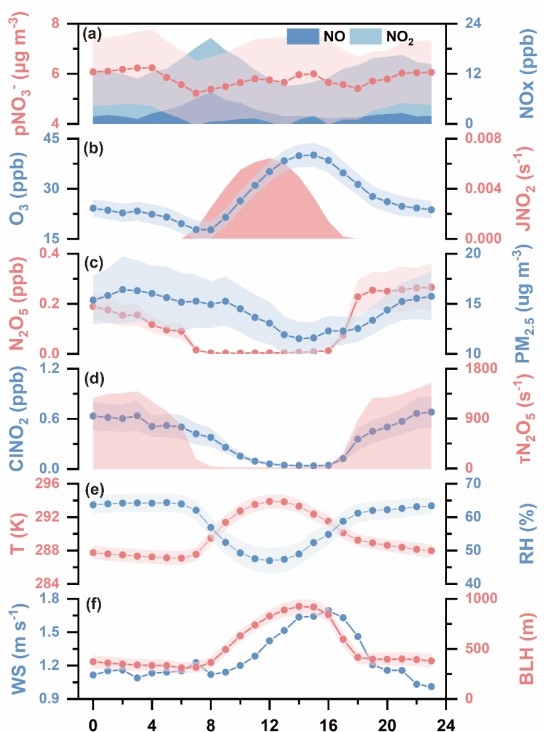


**Figure 1.** Diurnal variations of key parameters during the winter of 2022. The concentrations of $pNO_3^-$, $NOx$, $O_3$, $N_2O_5$, $PM_{2.5}$ and $ClNO_2$. The levels of the photolysis frequencies of $NO_2$ ($JNO_2$), ambient temperature (T), relative humidity (RH), the lifetime of $N_2O_5$ ($\tau N_2O_5$), wind speed (WS) and the boundary layer height (BLH). Shaded areas of $pNO_3^-$, $O_3$, $N_2O_5$, $PM_{2.5}$, $ClNO_2$, T, RH and BLH represent 95% confidence intervals.

**3.2 High contribution of $N_2O_5$ uptake to $pNO_3^-$ formation in $NO_2$-limited conditions.**

In view of the observed importance of daytime and nighttime $pNO_3^-$ formation, we further employed an observation-constrained model to quantify the potential formation pathways, including the gas-phase reaction of $OH + NO_2$ and heterogeneous $N_2O_5$ uptake. This model incorporated heterogeneous chemical mechanisms, with key heterogeneous parameters (e.g. the loss rate of $N_2O_5$ ($kN_2O_5$) and the production yield of $ClNO_2$ ($\varphi ClNO_2$)) obtained through simulation (See Methods for details). As shown in **Figure S7,** these simulated parameters exhibited good agreement with classical steady-state methods, demonstrating the model's capability to characterize heterogeneous uptake processes and thereby



effectively evaluate $pNO_3^-$ formation processes.
As illustrated in **Figure 2a**, the diurnal pattern of $pNO_3^-$ formation rates exhibited a classical
characteristic, with daytime dominated by gas-phase oxidation and nighttime dominated by $N_2O_5$ uptake.
The daytime $OH + NO_2$ reaction had a mean $pNO_3^-$ formation rate of 1.62 μg m$^{-3}$ h$^{-1}$, while the nighttime
$N_2O_5$ uptake pathway showed a formation rate of 1.18 μg m$^{-3}$ h$^{-1}$ (**Figure 2b-c**). For the whole day, $N_2O_5$
uptake contributed an average of 51.2% to $pNO_3^-$ formation, which was comparable to the contribution
of the $OH + NO_2$ pathway (**Figure 2d**). Notably, the partitioning coefficient for gas-phase oxidation
processes was assumed to be 1 in this study, meaning the contribution of $OH + NO_2$ represented an upper
limit and the actual contribution of $N_2O_5$ uptake should be even greater. To exclude year-specific effects,
we further analyzed the contributions of both pathways to $pNO_3^-$ formation during winters from 2019 to
2023. The results demonstrated that $N_2O_5$ uptake pathway consistently accounted for approximately half
of $pNO_3^-$ formation in the study area (**Figure 3a**), which was also consistent with the observed high
proportion of nighttime $pNO_3^-$ throughout the day (**Figure 3b**). Such a high contribution of $N_2O_5$ uptake
to $pNO_3^-$ is generally uncommon in urban areas. A study in urban Beijing showed that during non-
polluted periods, $N_2O_5$ uptake contributed only 18.9% to nitrate formation rates (Ma et al., 2023).
Similarly, the contributions of $N_2O_5$ uptake were 10%–38% and 4% in urban areas of the YRD (Sun et
al., 2022; Zhai et al., 2023; Zhang et al., 2023b) and PRD regions(Yang et al., 2022), respectively.
Previous studies have found that nocturnal $pNO_3^-$ formation via $N_2O_5$ uptake strongly depends on
the ratio of $NO_2$ to $O_3$ (Ma et al., 2023). This process is suppressed in the $O_3$-limited regime ($NO_2/O_3 >$
2) but enhanced in the $NO_2$-limited regime ($NO_2/O_3 \leq 1$). The COVID-19 lockdown period was a typical
example of this ratio dependence(Yan et al., 2023). In regions like Beijing, substantial reductions in NO*x*
emissions caused a shift in nocturnal $pNO_3^-$ formation from the $O_3$-limited to the $NO_2$-limited regime.
This shift resulted in elevated nighttime $O_3$ levels and a weakened NO titration effect, collectively
promoting $N_2O_5$ formation and subsequent $pNO_3^-$ formation. The sensitivity of $N_2O_5$ uptake to $NO_2$ and
$O_3$ during the campaign is presented in **Figure 3c-d**. The observed mean values of $NO_2/O_3$ (0.40) and the
probability distributions of $NO_2/O_3$ ratios both indicate that $N_2O_5$ uptake was in the $NO_2$-limited regime.
Based on $NO_2$ and $O_3$ observational data during 2015–2021 from the China National Environmental
Monitoring Centre[(Ma et al., 2023)], most key urban regions in China (e.g., the NCP, YRD, and Beijing) were
found to lie in the $O_3$-limited or transition regimes ($1 < NO_2/O_3 \leq 2$), whereas nocturnal $pNO_3^-$ formation
in southeastern China was distinctly in $NO_2$-limited regime. These results confirm that the dominant



pNO$_3^-$ formation mechanisms in our study area significantly differs from those in most urban areas of
China, which might be attributed to the dependence of N$_2$O$_5$ uptake on precursor NO$_2$ and O$_3$. In addition,
the dominance of N$_2$O$_5$ uptake in pNO$_3^-$ formation also occurred during haze pollution periods (Zhai et
al., 2023; Wang et al., 2017), where increased aerosol surface area under high particulate loadings created
favorable conditions for N$_2$O$_5$ heterogeneous reactions. Therefore, to evaluate the role of precursors, we
conducted a comprehensive analysis of the factors driving pNO$_3^-$ formation via N$_2$O$_5$ uptake.

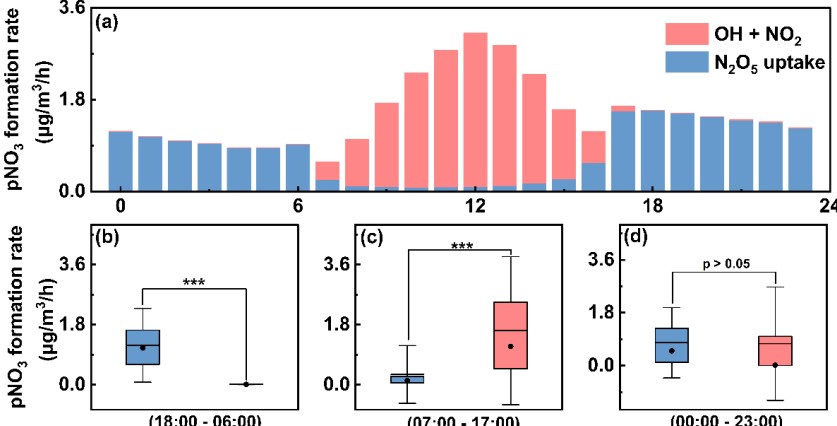


**Figure 2.** Simulated rates of key pNO$_3^-$ formation pathways obtained from the chemical box model
incorporating heterogeneous parameters. Diurnal formation rates of pNO$_3^-$ via the OH + NO$_2$ and N$_2$O$_5$
uptake pathways **(a)** and comparison of the two pathways during the nighttime **(b)**, daytime **(c)**, and the
whole day **(d)**. Note that the results in panel (a) represent the mean simulated formation rates over the
entire observation period. The box shows the 25th–75th percentiles with whiskers representing the 5th–
95th percentiles. The black line and dot inside the box represent the mean and median values, respectively.
Statistical significance was determined using pair-sample *t*-tests with *** indicating *p* < 0.001.



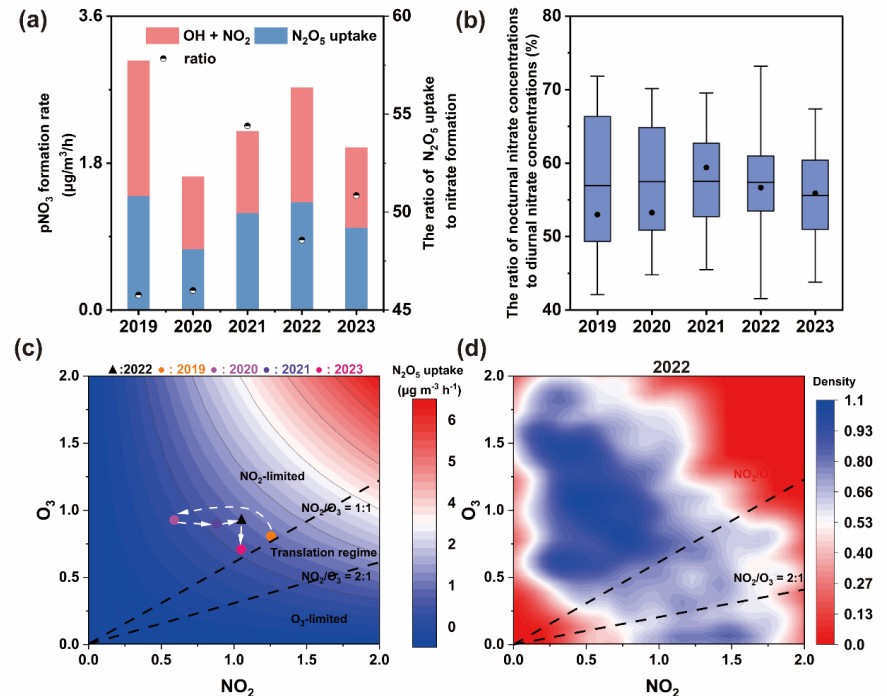

**Figure 3.** Inter-annual patterns of key $pNO_3^-$ formation pathways in urban Xiamen. The average $pNO_3^-$ formation rate from $OH + NO_2$ and $N_2O_5$ uptake **(a)**, and the average ratio of the sum of nocturnal $pNO_3^-$ concentrations to the sum of all-day $pNO_3^-$ concentration **(b)** in different winters from 2019 to 2023 based on the measured $pNO_3^-$ in $PM_{2.5}$. The sensitivity of nocturnal $N_2O_5$ uptake to $NO_2$ and $O_3$ from 2019 to 2023 **(c)**. And probability distribution of observed $NO_2/O_3$ at nighttime in winter 2022 **(d)**. The observed periods of different winters from 2019 to 2023 are summarized in **Table S3**. In panel (c), the black triangle indicates the base case of winter 2022, solid circles in different colors represent the average $NO_2$ to $O_3$ ratios in different years, and the predicted average formation rate of $N_2O_5$ uptake as the normalized emissions (average concentrations of $O_3$ and $NO_2$) varied between 0 to 2.

**3.3 Driving Factors of Nocturnal $N_2O_5$ Uptake.**

The $N_2O_5$ uptake rate is influenced by multiple factors including precursor levels, meteorological parameters, and heterogeneous reaction conditions (Ma et al., 2023; Chen et al., 2020; Chen et al., 2024). A machine learning method integrating these factors was employed to identify the key drivers of $N_2O_5$ uptake. The relative importance of each factor was evaluated by absolute SHAP values (**Figure 4a**), and



their impacts were elucidated by examining the relationships between individual factors and their
corresponding SHAP values (**Figure 4b-e** and **Figure S8**). Results showed that the nocturnal $NO_3$
formation rate ($P(NO_3)$), an integrated indicator of nocturnal atmospheric oxidation capacity (Wang et
al., 2021), was the most important factor for $N_2O_5$ uptake with the highest absolute SHAP value. The
steep slope of the positive correlation between $P(NO_3)$ and SHAP values indicated that $P(NO_3)$ strongly
enhanced $N_2O_5$ uptake. $P(NO_3)$ is primarily formed through the reaction between $NO_2$ and $O_3$ ($P(NO_3)$ =
$k_{NO_2+O_3}[NO_2][O_3]$), suggesting that $NO_2$ and $O_3$ mainly influenced $N_2O_5$ uptake by modulating $NO_3$
radical formation. Notably, the factor $logNO_2/O_3$ had relatively low importance, indicating
concentrations of precursors were more important than $NO_2/O_3$ ratio in determining the $N_2O_5$ uptake
under extremely $NO_2$-limited condition (mean $NO_2/O_3$ was 0.40). Furthermore, as shown in **Figure S8b**,
$logNO_2/O_3$ and its SHAP value shows a positive correlation when $logNO_2/O_3$ is less than 0. Under $NO_2$-
limited conditions ($logNO_2/O_3 < 0$, $NO_2/O_3 < 1$), $N_2O_5$ uptake was driven by the elevated $NO_2$.
Compared with $P(NO_3)$, other factors exhibited weaker effects on $N_2O_5$ uptake. $\varphi ClNO_2$ emerged
as the second most important factor and showed a negative correlation with SHAP values (**Figure 4c**),
illustrating that $ClNO_2$ formation inhibited $pNO_3^-$ formation. This inhibitory effect could be attributed to
high concentrations of Cl-containing particles ($0.94 \pm 1.11$ $\mu g$ $m^{-3}$) in the study area. Chloride-containing
aerosols promote $N_2O_5$ uptake to produce more $ClNO_2$ (as evidenced by the positive correlation between
$\varphi ClNO_2$ and chloride ions, **Figure S9**), while simultaneously reducing $pNO_3^-$ formation (R5).
Additionally, the nighttime produced $ClNO_2$ can undergo photolysis in following day to release Cl
radicals, which further promote $O_3$ formation. This indirect effect must be considered when formulating
control measures for particulate matter pollution. Interestingly, as shown in **Table S4** (Tham et al., 2016;
Wang et al., 2018; Yun et al., 2018; Morgan et al., 2015), although the simulated $kN_2O_5$ ($7.64\times10^{-3}$ $\pm$
$6.12\times10^{-3}$ $s^{-1}$) was higher than values reported in Beijing ($8.1\times10^{-4} – 1.42\times10^{-3}$ $s^{-1}$), Guangdong ($3.78\times10^{-3} – 9\times10^{-3}$ $s^{-1}$),
and UK ($9.3\times10^{-5} – 10^{-3}$ $s^{-1}$), $kN_2O_5$ exerted only a weak positive effect on $N_2O_5$ uptake
(**Figure 4d**). The large difference existing in importance of $P(NO_3)$ and $kN_2O_5$ indicated that the $N_2O_5$
uptake process was more limited by precursor levels rather than heterogeneous uptake conditions. Similar
phenomenon was also found in winter in urban Beijing and Northern Utah mountain basins (Mcduffie et
al., 2019; Chen et al., 2020). This situation is likely due to the favorable $N_2O_5$ uptake conditions during
winter, e.g., low temperature, high aerosol surface area, and elevated aerosol liquid content (Wang et al.,
2023a; Mcduffie et al., 2018b; Jia et al., 2020). The total concentrations of the observed VOCs (TVOCs)



showed a weak negative correlation with $N_2O_5$ uptake (**Figure 4e**), reflecting their indirect inhibition on
$N_2O_5$ formation by consuming $NO_3$ radicals. Moreover, we found that the effects of $\varphi ClNO_2$, $kN_2O_5$, and
TVOCs on $N_2O_5$ uptake were subject to $P(NO_3)$ levels (**Figure 5a-5c**). Specifically, the negative effect
of $\varphi ClNO_2$ and the positive effect of $kN_2O_5$ on $N_2O_5$ uptake became statistically significant when $P(NO_3)$
exceeded approximately 1.0 ppb $h^{-1}$ and 0.5 ppb $h^{-1}$, respectively. The negative correlation slope of
TVOCs versus $N_2O_5$ uptake intensified with increasing $P(NO_3)$ levels, indicating that the $N_2O_5$ removal
effect was enhanced through VOC-induced $NO_3$ depletion. These findings highlight the critical role of
precursor $NO_2$ and $O_3$ in nocturnal $pNO_3^-$ formation, demonstrating that these precursors mainly affect
this pathway by modulating $NO_3$ radical formation.



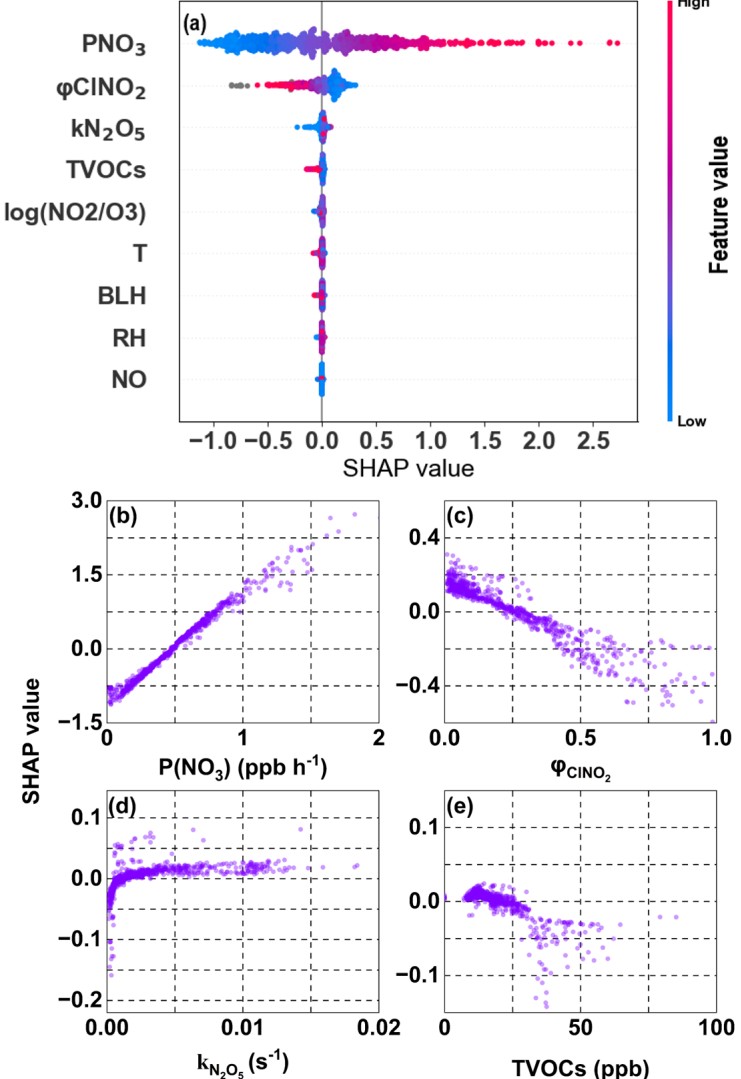

**Figure 4.** Feature importance **(a)** and the effects of key factors on $N_2O_5$ uptake **(b-e)** obtained by the

XGBoost-SHAP method. The relationships between SHAP values and major features: $P(NO_3)$ **(b)**,

$\varphi ClNO_2$**(c)**, $kN_2O_5$**(d)**, and TVOCs **(e)**. Feature importance ranking **(a)** is determined by mean absolute

SHAP values (descending order, top to bottom). Relationships between SHAP values and other factors

are shown in **Figure S8**.



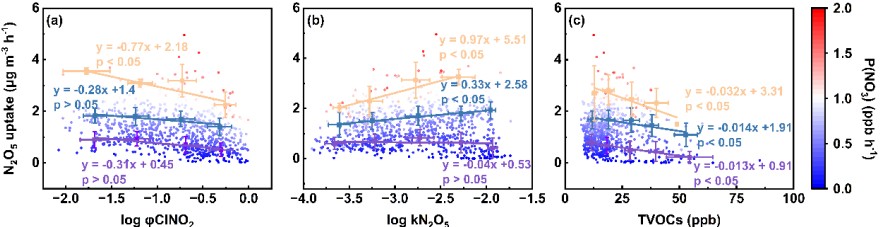

**Figure 5.** Relationships between $N_2O_5$ uptake and $\varphi ClNO_2$ **(a)**, $kN_2O_5$ **(b)**, and TVOCs **(c)** colored by $P(NO_3)$. Linear fit curves in purple, blue and orange represent the fitting results for $P(NO_3)$ in the ranges of 0–0.5 ppb h$^{-1}$, 0.5–1.0 ppb h$^{-1}$ and > 1.0 ppb h$^{-1}$, respectively.

**3.4 Optimal Mitigation Strategies of pNO$_3^-$ under High $N_2O_5$ Uptake.**

The above results revealed that pNO$_3^-$ formation through both the daytime OH + NO$_2$ reaction and nocturnal heterogeneous $N_2O_5$ uptake was closely linked to VOCs-NO$x$-O$_3$ chemistry (Yang et al., 2022). Using a multiphase box model, we systematically examined the responses of both pNO$_3^-$ and O$_3$ to varying NO$x$ and VOC emission scenarios. **Figure 6a** shows pNO$_3^-$ formation located in the transition regime of VOCs and NO$x$. The formation rate of pNO$_3^-$ (PNO$_3^-$) decreased with the reductions of VOCs and NO$x$, and this trend became more pronounced under aggressive NO$x$ reduction scenarios (**Figure 6c-d**). **Figure S10a-b** reveal that the mean response strength (RS, as defined in Methods) of PNO$_3^-$ to NO$x$ was 0.75, higher than that for VOCs (RS = 0.29), suggesting that NO$x$ reduction had a greater potential for pNO$_3^-$ mitigation compared to VOCs control. However, NO$x$ and VOCs reductions exerted different impacts on O$_3$ formation rate (PO$_3$). In our study area, PO$_3$ located in the VOC-limited regime (**Figure 6b**). We found that PO$_3$ declined with VOCs reduction but increased with NO$x$ reduction until NO$x$ dropped below 20% of the base (**Figure 5c-d**). Moreover, detailed results distinguishing daytime and nighttime major formation pathways of pNO$_3^-$ are presented in **Figure 6e-f** and **Fig. S10c-d**. For VOC reduction scenarios, both the OH + NO$_2$ reaction and N$_2$O$_5$ uptake pathways showed declining nitrate formation rates, with comparable RS of 0.11 and 0.18, respectively. This occurs because reduced VOCs concentrations decrease OH radical and O$_3$ concentrations, thereby suppressing pNO$_3^-$ formation via both pathways. In contrast, NO$x$ reduction yielded more complex behavior. The OH + NO$_2$ reaction rates remained nearly constant until NO$x$ dropped to 60% of the base. This stability arises because NO$x$ reduction diminishes the NO titration effect on O$_3$, thereby increasing OH radicals through O$_3$ photolysis.



The competing effects of NO$x$ reduction and OH enhancement led to an initial plateau in the OH + NO$_2$
reaction rate before its eventual decline. Differently, the N$_2$O$_5$ uptake rate decreased consistently and
significantly with NO$x$ abatement, exhibiting a high mean RS value of 0.61. This phenomenon was
closely associated with the NO$_2$-limited regime of N$_2$O$_5$ uptake in the study area. As shown in **Figure**
**S11,** the variation trends of PNO$_3^-$, P(O$_3$), OH + NO$_2$, and N$_2$O$_5$ uptake were consistent across all
VOCs/NO$x$ combinations, indicating that the results robustly reflect the response mechanisms to
precursor emission changes.
As mentioned above, VOC reduction proved effective yet limited in mitigating both pNO$_3^-$ and O$_3$,
the effectiveness of NO$x$ reduction exhibited significant regional and temporal variations. In China's
megacities, including PRD, YRD, and BTH regions, pNO$_3^-$ initially increased and then decreased in
response to the reduction of NO$x$ emissions (Li et al., 2021; Zhang et al., 2023b; Yang et al., 2022). Under
high-NO$x$ conditions, mild NO$x$ reduction would raise daytime OH and O$_3$ concentrations (Zhang et al.,
2023b), rendering OH (rather than NO$x$) the limiting factor for the OH + NO$_2$ reaction, which
consequently enhanced daytime pNO$_3^-$ formation. Additionally, as the season most susceptible to PM
pollution, wintertime N$_2$O$_5$ formation in these regions was in an O$_3$-limited or transition regime (Ma et
al., 2023), wherein the elevated daytime O$_3$ significantly enhanced NO$_3$ radical generation, thereby
promoting nocturnal N$_2$O$_5$ uptake and subsequent pNO$_3^-$ formation. Conversely, in NO$_2$-limited regions
(e.g., southeastern China), NO$x$ reduction showed limited impact on daytime pNO$_3^-$ formation via the
OH + NO$_2$ pathway but effectively suppressed nighttime pNO$_3^-$ formation via N$_2$O$_5$ uptake. This
approach concurrently reduced ClNO$_2$ formation from N$_2$O$_5$ heterogeneous processes, consequently
diminishing next-day Cl radical generation and its positive feedback on O$_3$ formation. Considering NO$x$
reduction during the daytime would cause O$_3$ formation and only a slight reduction in pNO$_3^-$, it is
preferable to regulate NO$x$ at night (18:00–06:00 the next day). Our findings demonstrate that in NO$_2$-
limited regions, targeted NO$x$ reduction can synergistically decrease both pNO$_3^-$ and O$_3$ concentrations,
highlighting the critical need to tailor mitigation strategies for different regions.



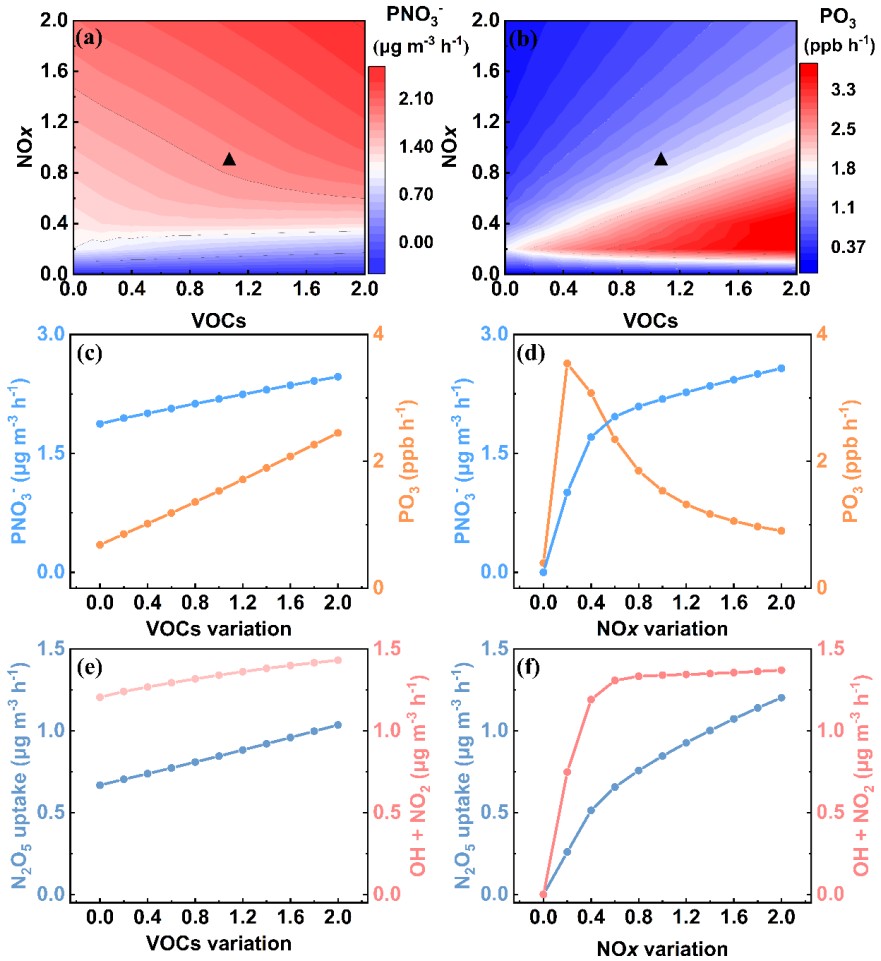

**Figure 6.** Results of multi-scenario simulations obtained from an observation-constrained box model. Isopleths of simulated PNO$_3^-$ **(a)** and PO$_3$ **(b)** with normalized VOCs and NO$x$**.** Simulated mean formation rates of pNO$_3^-$ and O$_3$ **(c, d)**, as well as pNO$_3^-$ formation rates via N$_2$O$_5$ uptake and OH + NO$_2$ **(e, f)** with normalized VOCs and NO$x$. The PNO$_3^-$ and PO$_3$ denote the formation rates of pNO$_3^-$ and O$_3$, respectively. The simulated results are daily mean values, and the black triangle indicates the base case for winter 2022. In addition, the results in panel c-f were obtained by maintaining either NO$x$ or VOCs at the base emission rate while varying the other.

**Conclusions and Implications**

Our observations revealed a bimodal diurnal pattern of pNO$_3^-$ in winter in urban Xiamen. The co-



occurrence of elevated nighttime $pNO_3^-$ levels with increased $N_2O_5$ implied a significant contribution of
$N_2O_5$ uptake to $pNO_3^-$ formation. Quantitative model analysis showed that $N_2O_5$ uptake contributed 51.2%
of the total daily $pNO_3^-$, which was comparable to the $OH + NO_2$ reaction. This high contribution of $N_2O_5$
uptake is not commonly observed across Chinese cities. Comparative analysis among different cities
suggests that this phenomenon is likely associated with $NO_2$-limited conditions for $N_2O_5$ uptake in our
study area. Machine learning results further demonstrated that $N_2O_5$ uptake was driven by nocturnal
atmospheric oxidation capacity ($PNO_3$) rather than heterogeneous uptake conditions. The underlying
mechanism is that the weakened $NOx$ titration effects lead to nighttime $O_3$ accumulation, which promotes
$NO_3$ radical generation and consequently enhances $N_2O_5$ and $pNO_3^-$ formation. The joint response of
$pNO_3^-$ and $O_3$ to various $NOx$ and VOCs emission scenarios indicated that $pNO_3^-$ was more sensitive to
$NOx$ reduction than to VOCs reduction. However, mild $NOx$ reduction showed limited effectiveness in
reducing daytime $pNO_3^-$ while simultaneously increasing $O_3$ concentrations. Our findings suggest that
$NOx$ reduction is more effective when implemented during nighttime, particularly in regions where $N_2O_5$
formation is $NO_2$-limited. This approach can effectively control $pNO_3^-$ formation by suppressing
nocturnal $NO_3$ radical generation and consequently inhibiting $N_2O_5$ uptake, while simultaneously
alleviate $O_3$ pollution by reducing $ClNO_2$ formation. With continuous $NOx$ and VOCs emission
reductions and renewable energy adoption in China, urban areas are transitioning from $NOx$-saturated to
$NOx$-limited conditions, potentially increasing the importance of $N_2O_5$ uptake. In this context,
comprehensive assessment of $NOx$ reduction impacts on urban $pNO_3^-$ and $O_3$ pollution, along with the
development of region-specific mitigation strategies, becomes critically important.

**Data Availability**
The dataset for this paper can be accessed at https://doi.org/10.6084/m9.figshare.29670629 (Lin et al.,

408  2025).


**Code Availability**
Data analysis methods are available from the authors upon request.

**Acknowledgements**
This work was funded by the National Natural Science Foundation of China (U22A20578), the guiding



project of seizing the commanding heights of "self-purifying city" (IUE-CERAE-202402), the National
Key Research and Development Program (2022YFC3700304), STS Plan Supporting Project of the
Chinese Academy of Sciences in Fujian Province (2023T3013), and Xiamen Atmospheric Environment
Observation and Research Station of Fujian Province.

**Author Contribution**
Z.L. contributed to the methodology, data curation, software, analysis and writing of the original draft.
L.X. and J.C. contributed to the conceptualization, investigation, data curation, reviewing and editing the
text, supervision, and funding acquisition. C.Y., X.J., K.Z., F.Z., G.C., L.L., C.Y., Y.C., and Z.C. provided
useful advice and revised the manuscript.

**Competing interests**
The authors declare no competing interests.

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

J., and He, H.: Formation of Nitrate in the Residual Layer of Beijing: Pathways Evaluation and
Contributions to the Ground Level, Environmental Science & Technology. 59, 9699-9708,
https://doi.org/10.1021/acs.est.5c02981, 2025.
Ma, P. K., Quan, J. N., Dou, Y. J., Pan, Y. B., Liao, Z. H., Cheng, Z. G., Jia, X. C., Wang, Q. Q., Zhan, J.
L., Ma, W., Zheng, F. X., Wang, Y. Z., Zhang, Y. S., Hua, C. J., Yan, C., Kulmala, M., Liu, Y. A., Huang,
X., Yuan, B., Brown, S. S., and Liu, Y. C.: Regime-Dependence of Nocturnal Nitrate Formation via N2O5
Hydrolysis and Its Implication for Mitigating Nitrate Pollution, Geophysical Research Letters. 50,
https://doi.org/10.1029/2023gl106183, 2023.
Mao, J. Y., Yan, F. H., Zheng, L. M., You, Y. C., Wang, W. W., Jia, S. G., Liao, W. H., Wang, X. M., and
Chen, W. H.: Ozone control strategies for local formation- and regional transport-dominant scenarios in
a manufacturing city in southern China, Science of the Total Environment. 813,
https://doi.org/10.1016/j.scitotenv.2021.151883, 2022.
McDuffie, E. E., Womack, C. C., Fibiger, D. L., Dube, W. P., Franchin, A., Middlebrook, A. M.,
Goldberger, L., Lee, B., Thornton, J. A., Moravek, A., Murphy, J. G., Baasandorj, M., and Brown, S. S.:
On the contribution of nocturnal heterogeneous reactive nitrogen chemistry to particulate matter
formation during wintertime pollution events in Northern Utah, Atmospheric Chemistry and Physics. 19,
9287-9308, https://doi.org/10.5194/acp-19-9287-2019, 2019.
McDuffie, E. E., Fibiger, D. L., Dubé, W. P., Hilfiker, F. L., Lee, B. H., Jaeglé, L., Guo, H. Y., Weber, R.
J., Reeves, J. M., Weinheimer, A. J., Schroder, J. C., Campuzano-Jost, P., Jimenez, J. L., Dibb, J. E.,
Veres, P., Ebben, C., Sparks, T. L., Wooldridge, P. J., Cohen, R. C., Campos, T., Hall, S. R., Ullmann, K.,
Roberts, J. M., Thornton, J. A., and Brown, S. S.: ClNO2 Yields From Aircraft Measurements During the
2015 WINTER Campaign and Critical Evaluation of the Current Parameterization, Journal of
Geophysical Research-Atmospheres. 123, 12994-13015, https://doi.org/10.1029/2018jd029358, 2018a.
McDuffie, E. E., Fibiger, D. L., Dubé, W. P., Lopez-Hilfiker, F., Lee, B. H., Thornton, J. A., Shah, V.,
Jaeglé, L., Guo, H. Y., Weber, R. J., Reeves, J. M., Weinheimer, A. J., Schroder, J. C., Campuzano-Jost,
P., Jimenez, J. L., Dibb, J. E., Veres, P., Ebben, C., Sparks, T. L., Wooldridge, P. J., Cohen, R. C.,
Hornbrook, R. S., Apel, E. C., Campos, T., Hall, S. R., Ullmann, K., and Brown, S. S.: Heterogeneous
N2O5 Uptake During Winter: Aircraft Measurements During the 2015 WINTER Campaign and Critical
Evaluation of Current Parameterizations, Journal of Geophysical Research-Atmospheres. 123, 4345-
4372, https://doi.org/10.1002/2018jd028336, 2018b.
Morgan, W. T., Ouyang, B., Allan, J. D., Aruffo, E., Di Carlo, P., Kennedy, O. J., Lowe, D., Flynn, M. J.,
Rosenberg, P. D., Williams, P. I., Jones, R., McFiggans, G. B., and Coe, H.: Influence of aerosol chemical
composition on N2O5 uptake: airborne regional measurements in northwestern Europe, Atmospheric



Chemistry and Physics. 15, 973-990, https://doi.org/10.5194/acp-15-973-2015, 2015.
Niu, Y. B., Zhu, B., He, L. Y., Wang, Z., Lin, X. Y., Tang, M. X., and Huang, X. F.: Fast Nocturnal
Heterogeneous Chemistry in a Coastal Background Atmosphere and Its Implications for Daytime
Photochemistry,        Journal        of        Geophysical        Research-Atmospheres.        127,
https://doi.org/10.1029/2022jd036716, 2022.
Requia, W. J., Di, Q., Silvern, R., Kelly, J. T., Koutrakis, P., Mickley, L. J., Sulprizio, M. P., Amini, H.,
Shi, L. H., and Schwartz, J.: An Ensemble Learning Approach for Estimating High Spatiotemporal
Resolution of Ground-Level Ozone in the Contiguous United States, Environmental Science &
Technology. 54, 11037-11047, https://doi.org/10.1021/acs.est.0c01791, 2020.
Seinfeld, J. H.: URBAN AIR-POLLUTION - STATE OF THE SCIENCE, Science. 243, 745-752,
https://doi.org/10.1126/science.243.4892.745, 1989.
Sun, J. J., Qin, M. M., Xie, X. D., Fu, W. X., Qin, Y., Sheng, L., Li, L., Li, J. Y., Sulaymon, I. D., Jiang,
L., Huang, L., Yu, X. N., and Hu, J. L.: Seasonal modeling analysis of nitrate formation pathways in
Yangtze River Delta region, China, Atmospheric Chemistry and Physics. 22, 12629-12646,
https://doi.org/10.5194/acp-22-12629-2022, 2022.
Thaler, R. D., Mielke, L. H., and Osthoff, H. D.: Quantification of Nitryl Chloride at Part Per Trillion
Mixing Ratios by Thermal Dissociation Cavity Ring-Down Spectroscopy, Analytical Chemistry. 83,
2761-2766, https://doi.org/10.1021/ac200055z, 2011.
Tham, Y. J., Wang, Z., Li, Q. Y., Wang, W. H., Wang, X. F., Lu, K. D., Ma, N., Yan, C., Kecorius, S.,
Wiedensohler, A., Zhang, Y. H., and Wang, T.: Heterogeneous $N_2O_5$ uptake coefficient and production
yield of $ClNO_2$ in polluted northern China: roles of aerosol water content and chemical composition,
Atmospheric Chemistry and Physics. 18, 13155-13171, https://doi.org/10.5194/acp-18-13155-2018,
562  2018.

Tham, Y. J., Wang, Z., Li, Q. Y., Yun, H., Wang, W. H., Wang, X. F., Xue, L. K., Lu, K. D., Ma, N., Bohn,
B., Li, X., Kecorius, S., Gröss, J., Shao, M., Wiedensohler, A., Zhang, Y. H., and Wang, T.: Significant
concentrations of nitryl chloride sustained in the morning: investigations of the causes and impacts on
ozone production in a polluted region of northern China, Atmospheric Chemistry and Physics. 16, 14959-
14977, https://doi.org/10.5194/acp-16-14959-2016, 2016.
Wagner, N. L., Riedel, T. P., Young, C. J., Bahreini, R., Brock, C. A., Dubé, W. P., Kim, S., Middlebrook,
A. M., Öztürk, F., Roberts, J. M., Russo, R., Sive, B., Swarthout, R., Thornton, J. A., VandenBoer, T. C.,
Zhou, Y., and Brown, S. S.: $N_2O_5$ uptake coefficients and nocturnal $NO_2$ removal rates determined from
ambient wintertime measurements, Journal of Geophysical Research-Atmospheres. 118, 9331-9350,
https://doi.org/10.1002/jgrd.50653, 2013.
Wang, H. C., Lu, K. D., Chen, S. Y., Li, X., Zeng, L. M., Hu, M., and Zhang, Y. H.: Characterizing nitrate
radical budget trends in Beijing during 2013-2019, Science of the Total Environment. 795,
https://doi.org/10.1016/j.scitotenv.2021.148869, 2021.
Wang, H. C., Wang, H. L., Lu, X., Lu, K. D., Zhang, L., Tham, Y. J., Shi, Z. B., Aikin, K., Fan, S. J.,
Brown, S. S., and Zhang, Y. H.: Increased night-time oxidation over China despite widespread decrease
across the globe, Nature Geoscience. 16, 217-+, https://doi.org/10.1038/s41561-022-01122-x, 2023a.
Wang, H. C., Peng, C., Wang, X., Lou, S. R., Lu, K. D., Gan, G. C., Jia, X. H., Chen, X. R., Chen, J.,
Wang, H. L., Fan, S. J., Wang, X. M., and Tang, M. J.: N2O5 uptake onto saline mineral dust: a potential
missing source of tropospheric $ClNO_2$ in inland China, Atmospheric Chemistry and Physics. 22, 1845-
1859, https://doi.org/10.5194/acp-22-1845-2022, 2022a.
Wang, H. C., Lu, K. D., Guo, S., Wu, Z. J., Shang, D. J., Tan, Z. F., Wang, Y. J., Le Breton, M., Lou, S.



R., Tang, M. J., Wu, Y. S., Zhu, W. F., Zheng, J., Zeng, L. M., Hallquist, M., Hu, M., and Zhang, Y. H.:
Efficient $N_2O_5$ uptake and $NO_3$ oxidation in the outflow of urban Beijing, Atmospheric Chemistry and
Physics. 18, 9705-9721, https://doi.org/10.5194/acp-18-9705-2018, 2018.
Wang, H. C., Lu, K. D., Chen, X. R., Zhu, Q. D., Chen, Q., Guo, S., Jiang, M. Q., Li, X., Shang, D. J.,
Tan, Z. F., Wu, Y. S., Wu, Z. J., Zou, Q., Zheng, Y., Zeng, L. M., Zhu, T., Hu, M., and Zhang, Y. H.: High
$N_2O_5$ Concentrations Observed in Urban Beijing: Implications of a Large Nitrate Formation Pathway,
Environmental Science & Technology Letters. 4, 416-420, https://doi.org/10.1021/acs.estlett.7b00341,
591     2017.
Wang, H. C., Yuan, B., Zheng, E., Zhang, X. X., Wang, J., Lu, K. D., Ye, C. S., Yang, L., Huang, S., Hu,
W. W., Yang, S. X., Peng, Y. W., Qi, J. P., Wang, S. H., He, X. J., Chen, Y. B., Li, T. G., Wang, W. J.,
Huangfu, Y. B., Li, X. B., Cai, M. F., Wang, X. M., and Shao, M.: Formation and impacts of nitryl chloride
in Pearl River Delta, Atmospheric Chemistry and Physics. 22, 14837-14858, https://doi.org/10.5194/acp-
22-14837-2022, 2022b.
Wang, W. J., Li, X., Cheng, Y. F., Parrish, D. D., Ni, R. J., Tan, Z. F., Liu, Y., Lu, S. H., Wu, Y. S., Chen,
S. Y., Lu, K. D., Hu, M., Zeng, L. M., Shao, M., Huang, C., Tian, X. D., Leung, K. M., Chen, L. F., Fan,
M., Zhang, Q., Rohrer, F., Wahner, A., Pöschl, U., Su, H., and Zhang, Y. H.: Ozone pollution mitigation
strategy informed by long-term trends of atmospheric oxidation capacity, Nature Geoscience. 16, 1080-
1081, https://doi.org/10.1038/s41561-023-01334-9, 2023b.
Wang, Y. H., Gao, W. K., Wang, S., Song, T., Gong, Z. Y., Ji, D. S., Wang, L. L., Liu, Z. R., Tang, G. Q.,
Huo, Y. F., Tian, S. L., Li, J. Y., Li, M. G., Yang, Y., Chu, B. W., Petäjä, T., Kerminen, V. M., He, H., Hao,
J. M., Kulmala, M., Wang, Y. S., and Zhang, Y. H.: Contrasting trends of $PM_{2.5}$ and surface-ozone
concentrations in China from 2013 to 2017, National Science Review. 7, 1331-1339,
https://doi.org/10.1093/nsr/nwaa032, 2020.
Wang, Y. R., Yang, X. Y., Wu, K., Mei, H., De Smedt, I., Wang, S. G., Fan, J., Lyu, S., and He, C.: Long-
term trends of ozone and precursors from 2013 to 2020 in a megacity (Chengdu), China: Evidence of
changing      emissions      and      chemistry,      Atmospheric      Research.      278,
https://doi.org/10.1016/j.atmosres.2022.106309, 2022c.
Wang, Y. T., Zhao, Y., Liu, Y. M., Jiang, Y. Q., Zheng, B., Xing, J., Liu, Y., Wang, S., and Nielsen, C. P.:
Sustained emission reductions have restrained the ozone pollution over China, Nature Geoscience. 16,
967-+, https://doi.org/10.1038/s41561-023-01284-2, 2023c.
Wen, L., Xue, L. K., Wang, X. F., Xu, C. H., Chen, T. S., Yang, L. X., Wang, T., Zhang, Q. Z., and Wang,
W. X.: Summertime fine particulate nitrate pollution in the North China Plain: increasing trends,
formation mechanisms and implications for control policy, Atmospheric Chemistry and Physics. 18,
11261-11275, https://doi.org/10.5194/acp-18-11261-2018, 2018.
Wolfe, G. M., Marvin, M. R., Roberts, S. J., Travis, K. R., and Liao, J.: The Framework for 0-D
Atmospheric Modeling (F0AM) v3.1, Geoscientific Model Development. 9, 3309-3319,
https://doi.org/10.5194/gmd-9-3309-2016, 2016.
Xie, X. D., Hu, J. L., Qin, M. M., Guo, S., Hu, M., Wang, H. L., Lou, S. R., Li, J. Y., Sun, J. J., Li, X.,
Sheng, L., Zhu, J. L., Chen, G. Y., Yin, J. J., Fu, W. X., Huang, C., and Zhang, Y. H.: Modeling particulate
nitrate in China: Current findings and future directions, Environment International. 166,
https://doi.org/10.1016/j.envint.2022.107369, 2022.
Xing, J., Ding, D., Wang, S. X., Zhao, B., Jang, C., Wu, W. J., Zhang, F. F., Zhu, Y., and Hao, J. M.:
Quantification of the enhanced effectiveness of $NO_x$ control from simultaneous reductions of VOC and
NH3 for reducing air pollution in the Beijing-Tianjin-Hebei region, China, Atmospheric Chemistry and



Physics. 18, 7799-7814, https://doi.org/10.5194/acp-18-7799-2018, 2018.
Yan, C., Tham, Y. J., Nie, W., Xia, M., Wang, H. C., Guo, Y. S., Ma, W., Zhan, J. L., Hua, C. J., Li, Y. Y.,
Deng, C. J., Li, Y. R., Zheng, F. X., Chen, X., Li, Q. Y., Zhang, G., Mahajan, A. S., Cuevas, C. A., Huang,
D. D., Wang, Z., Sun, Y. L., Saiz-Lopez, A., Bianchi, F., Kerminen, V. M., Worsnop, D. R., Donahue, N.
M., Jiang, J. K., Liu, Y. C., Ding, A. J., and Kulmala, M.: Increasing contribution of nighttime nitrogen
chemistry to wintertime haze formation in Beijing observed during COVID-19 lockdowns, Nature
Geoscience. 16, 975-+, https://doi.org/10.1038/s41561-023-01285-1, 2023.
Yang, C., Dong, H. S., Chen, Y. P., Xu, L. L., Chen, G. J., Fan, X. L., Wang, Y. H., Tham, Y. J., Lin, Z.
Y., Li, M. R., Hong, Y. W., and Chen, J. S.: New Insights on the Formation of Nucleation Mode Particles
in a Coastal City Based on a Machine Learning Approach, Environmental Science & Technology. 58,
1187-1198, https://doi.org/10.1021/acs.est.3c07042, 2023.
Yang, S. X., Yuan, B., Peng, Y. W., Huang, S., Chen, W., Hu, W. W., Pei, C. L., Zhou, J., Parrish, D. D.,
Wang, W. J., He, X. J., Cheng, C. L., Li, X. B., Yang, X. Y., Song, Y., Wang, H. C., Qi, J. P., Wang, B. L.,
Wang, C., Wang, C. M., Wang, Z. L., Li, T. G., Zheng, E., Wang, S. H., Wu, C. H., Cai, M. F., Ye, C. S.,
Song, W., Cheng, P., Chen, D. H., Wang, X. M., Zhang, Z. Y., Wang, X. M., Zheng, J. Y., and Shao, M.:
The formation and mitigation of nitrate pollution: comparison between urban and suburban environments,
Atmospheric Chemistry and Physics. 22, 4539-4556, https://doi.org/10.5194/acp-22-4539-2022, 2022.
Yu, C., Wang, Z., Xia, M., Fu, X., Wang, W. H., Tham, Y. J., Chen, T. S., Zheng, P. G., Li, H. Y., Shan,
Y., Wang, X. F., Xue, L. K., Zhou, Y., Yue, D. L., Ou, Y. B., Gao, J., Lu, K. D., Brown, S. S., Zhang, Y.
H., and Wang, T.: Heterogeneous N2O5 reactions on atmospheric aerosols at four Chinese sites:
improving model representation of uptake parameters, Atmospheric Chemistry and Physics. 20, 4367-
4378, https://doi.org/10.5194/acp-20-4367-2020, 2020.
Yun, H., Wang, W. H., Wang, T., Xia, M., Yu, C., Wang, Z., Poon, S. C. N., Yue, D. L., and Zhou, Y.:
Nitrate formation from heterogeneous uptake of dinitrogen pentoxide during a severe winter haze in
southern China, Atmospheric Chemistry and Physics. 18, 17515-17527, https://doi.org/10.5194/acp-18-

653    17515-2018, 2018.

Zhai, S. X., Jacob, D. J., Wang, X., Liu, Z. R., Wen, T. X., Shah, V., Li, K., Moch, J. M., Bates, K. H.,
Song, S. J., Shen, L., Zhang, Y. Z., Luo, G., Yu, F. Q., Sun, Y. L., Wang, L. T., Qi, M. Y., Tao, J., Gui, K.,
Xu, H. H., Zhang, Q., Zhao, T. L., Wang, Y. S., Lee, H. C., Choi, H., and Liao, H.: Control of particulate
nitrate air pollution in China, Nature Geoscience. 14, 389-+, https://doi.org/10.1038/s41561-021-00726-
z, 2021.
Zhai, T. Y., Lu, K. D., Wang, H. C., Lou, S. R., Chen, X. R., Hu, R. Z., and Zhang, Y. H.: Elucidate the
formation mechanism of particulate nitrate based on direct radical observations in the Yangtze River
Delta summer 2019, Atmospheric Chemistry and Physics. 23, 2379-2391, https://doi.org/10.5194/acp-

662    23-2379-2023, 2023.

Zhang, R., Han, Y. H., Shi, A. J., Sun, X. S., Yan, X., Huang, Y. H., and Wang, Y.: Characteristics of
ambient ammonia and its effects on particulate ammonium in winter of urban Beijing, China,
Environmental Science and Pollution Research. 28, 62828-62838, https://doi.org/10.1007/s11356-021-
14108-w, 2021.
Zhang, X., Ma, Q., Chu, W. H., Ning, M., Liu, X. Q., Xiao, F. J., Cai, N. N., Wu, Z. J., and Yan, G.:
Identify the key emission sources for mitigating ozone pollution: A case study of urban area in the
Yangtze River Delta region, China, Science of the Total Environment. 892,
https://doi.org/10.1016/j.scitotenv.2023.164703, 2023a.
Zhang, Y., Lei, R., Cui, S., Wang, H., Chen, M., and Ge, X.: Spatiotemporal trends and impact factors of



PM$_{2.5}$ and O$_3$ pollution in major cities in China during 2015-2020, Chinese Science Bulletin. 67, 2029-
673    2042, 2022.

Zhang, Y. N., Wang, H. L., Huang, L. B., Qiao, L. P., Zhou, M., Mu, J. S., Wu, C., Zhu, Y. J., Shen, H.
Q., Huang, C., Wang, G. H., Wang, T., Wang, W. X., and Xue, L. K.: Double-Edged Role of VOCs
Reduction in Nitrate Formation: Insights from Observations during the China International Import Expo
2018, Environmental Science & Technology. 57, 15979-15989, https://doi.org/10.1021/acs.est.3c04629,
2023b.
Zhao, S. P., Yin, D. Y., Yu, Y., Kang, S. C., Qin, D. H., and Dong, L. X.: PM$_{2.5}$ and O$_3$ pollution during
2015-2019 over 367 Chinese cities: Spatiotemporal variations, meteorological and topographical impacts,
Environmental Pollution. 264, https://doi.org/10.1016/j.envpol.2020.114694, 2020.
Zhao, X. X., Zhao, X. J., Liu, P. F., Chen, D., Zhang, C. L., Xue, C. Y., Liu, J. F., Xu, J., and Mu, Y. J.:
Transport Pathways of Nitrate Formed from Nocturnal N$_2$O$_5$ Hydrolysis Aloft to the Ground Level in
Winter North China Plain, Environmental Science & Technology.
https://doi.org/10.1021/acs.est.3c00086, 2023.
Zhou, M., Nie, W., Qiao, L. P., Huang, D. D., Zhu, S. H., Lou, S. R., Wang, H. L., Wang, Q., Tao, S. K.,
Sun, P., Liu, Y. W., Xu, Z., An, J. Y., Yan, R. S., Su, H., Huang, C., Ding, A. J., and Chen, C. H.: Elevated
Formation of Particulate Nitrate From N$_2$O$_5$ Hydrolysis in the Yangtze River Delta Region From 2011 to
2019, Geophysical Research Letters. 49, https://doi.org/10.1029/2021gl097393, 2022.
Zong, Z., Tian, C. G., Sun, Z. Y., Tan, Y., Shi, Y. J., Liu, X. H., Li, J., Fang, Y. T., Chen, Y. J., Ma, Y. H.,
Gao, H. W., Zhang, G., and Wang, T.: Long-Term Evolution of Particulate Nitrate Pollution in North
China: Isotopic Evidence From 10 Offshore Cruises in the Bohai Sea From 2014 to 2019, Journal of
Geophysical Research-Atmospheres. 127, https://doi.org/10.1029/2022jd036567, 2022.