# Peer review of "Measurement report: High contribution of N2O5 uptake"

_EGUsphere, 2025_

## Author Comment (AC1)

Reply to comments on "Measurement report: High contribution of N2O5 uptake to particulate nitrate formation in NO2-limited urban areas" by Lin et al.

We would like to thank the editor and reviewers for their efforts in handling, reading, and critically reviewing our manuscript, which have helped us to further improve our manuscript. The comments on our paper are carefully addressed.

**General Comments:**

This study investigates the features of pNO3- production at a typical site in southeastern China. The nighttime N2O5 uptake was found to be efficient enough and play an increasing role in pNO3- formation. Multiple methods are applied to demonstrating the dominant effect of PNO3 or precursor concentrations, in N2O5 uptake process. The manuscript is well organized. I recommend acceptance after carefully addressing the following concerns when revising this manuscript.

Response: We are grateful for your thoughtful comments on the manuscript and we have made careful revisions accordingly. Our point-to-point responses to each comment are as follows (reviewer's comments are in black font, our responses are in blue font and our revisions in the manuscript are italic font).

1. Method 2.1: I suggest more description of instrument deployment at least for I-Tof-CIMS in main text. It would be better to briefly introduce the sampling setup, calibration frequency and the variation of calibration factor, such that the reasonableness of measurements could be readily assessed.

Response: Thank you for this suggestion. We have added the detailed operation and calibration procedure for the I-TOF-CIMS in the revised manuscript (line 112-127), as follows:

"A nearly 2-meter long perfluoroalkoxy (PFA) tube with a 1/4-inch inner diameter was used for sampling. The total sampling flow rate was set as 10 standard liters per minute (SLPM), of which only 2SLPM was diverted to the CIMS. A nitrogen (N2) flow (99.999%, 2.7 SLPM), carrying methyl iodide (CH3I) vapor released from a heated permeation tube, passed through a soft X-ray source (Tofwerk AG, P-type) to generate reagent ions I. The I was combined with the target gas in an ion molecule reaction (IMR) chamber and then detected by the ToF-CIMS. Ambient  $N_2O_5$  and ClNO2 were detected as the  $I(N_2O_5)^-$  and  $I(ClNO_2)^-$  clusters at 235 and 208 m/z. The detailed calibration procedures of N2O5 and ClNO2 are described in **Text S2**, following established methods (Wang et al., 2022b; Wang et al., 2022a; Thaler et al., 2011). Briefly,  $N_2O_5$  was generated from the reaction between  $O_3$  and excessive  $NO_2$ , while  $ClNO_2$  was synthesized via the reaction of  $Cl_2$  (6 ppm in  $N_2$ ) with a moist mixture of NaNO2 and NaCl. The calibration curves for N2O5 and ClNO2 at different RH are shown in Figure S2, with mean sensitivities of  $0.110 \pm 0.063$  and  $0.055 \pm 0.018$ ncps/ppb, respectively. The instrument background was determined by introducing dry  $N_2$  into the inlet for 20 min. Based on three times the standard deviation (3 $\sigma$ ) of the background signal, the typical 1-minute detection limits for N2O5 and ClNO2 were estimated to be 1.3 and 0.61 ppt, respectively."

2. Method 2.2: The algorithm proposed by Wagner 2013 should be termed as iterative box model instead of interactive box model.

Response: Thank you for the note. We have corrected it in the revised manuscript (line 130,132 and 139).

3. Line 212: Suggest clearly stating what simulate parameters exhibited good agreement here.

Response: As you suggested, we have rewritten the sentence in the revised manuscript (line 225-228) as follows:

"As shown in **Figure S7**, the simulated  $kN_2O_5$  and  $\varphi ClNO_2$  exhibited good agreement with the classical steady-state method ( $R^2 = 0.76$  and 0.73, respectively), demonstrating the model's capability to characterize heterogeneous uptake processes and thereby effectively evaluate  $pNO_3^-$  formation processes."

4. Line 218-230: A direct comparison of pNO3- formation rate is also helpful to indicate the characteristics of pNO3- formation at this site.

Response: Thank you for the suggestion. We have added a direct comparison of pNO3- formation rate in the revised manuscript (line 234-238) as follows:

"To exclude year-specific effects, we further analyzed pNO3- formation during the winters from 2019 to 2023. The results revealed that the pNO3- formation rates via N2O5 uptake  $(0.75 - 1.40 \,\mu g \, m^{-3} \, h^{-1})$  were comparable to those from the OH + NO2 reaction  $(0.88 - 1.66 \,\mu g \, m^{-3} \, h^{-1})$ ; **Figure 3a**), with the N2O5 uptake pathway consistently

accounting for approximately half of the total  $pNO_3^-$  formation in the study area (Figure 3b)."

5. Line 241: Please revise the text font of citation.

Response: Thank you for the note. We have corrected it in the revised manuscript (line 253).

6. Line 307-314: The SHAP of TVOC exhibit minor impact on N2O5 uptake and correlate not so well with its concentration. Replacing TVOCs with specific VOC species, such as monoterpene and styrene, could provide better correlation of this feature.

Response: Thank you for the comment. We agree that specific VOCs species influence N2O5 formation due to the higher reactivity of NO3 toward them. In this work, the loss of N2O5 was calculated using eqs. R1-R2 (eqs. S3-S4 in the supplementary material), where kN2O5 represents the rate of N2O5 uptake, and kNO3/Keq[NO2] corresponds to the indirect chemical loss of N2O5 through NO3 chemistry. As shown in Table S6, the reaction rate of kNO3/Keq[NO2] was calculated to be 0.000136 s-1, which is much smaller than that of kN2O5 (0.00764 s-1). This indicates that the loss of N2O5 through the consumption of its precursors NO3 by VOCs is relatively limited compared to its direct uptake. Considering this finding and the risk of model overfitting when including too many variables, we used TVOCs as a simplified indicator to represent the effect of VOCs on pNO3- formation via N2O5 uptake. As a result, the low SHAP value of TVOCs

is consistent with their limited influence on N2O5 removal, as determined by our calculations. We have included a detailed discussion in the supplementary material (line 98-104) and provided corresponding explanations in the revised manuscript (line 319-324).

$$\tau_{N_2O_5} = \frac{[N_2O_5]}{K_1(T)[NO_2][O_3]} \tag{R1}$$

$$(\tau_{N_2O_5})^{-1} = kN_2O_5 + \frac{k_{NO_3}}{K_{eq}[NO_2]}$$
 (R2)

The corresponding explanations in the main text are as follows:

"The total concentrations of the observed VOCs (TVOCs) showed a weak negative correlation with N2O5 uptake (**Figure 4e**). Similar to existing research (Hu et al., 2023), specific VOC species, such as styrene, 2-butene, and isoprene, can readily consume NO3 radicals (**Figure S10**), thereby inhibiting N2O5 formation. However, the loss of N2O5 through the reaction between VOCs and NO3 was relatively limited compared to its direct uptake, as determined by our calculations (Text S4), which supported the SHAP analysis."

7. Section 3.4: The response of pNO3- and O3 production rate to precursors is well investigated, while I figure out two confusing points in discussion part. First, the production of O3 was clearly proved to be VOC-limited, resulting in effective mitigation on O3 by reducing VOC. Meanwhile, the pNO3- production also shows larger sensitivity to VOC variation. However, the authors claim that the effect of VOC reduction is limited in mitigating both pNO3- and O3, which is confusing. Second, the title of this manuscript, a NO2-limited region, seems contradictory to the finding of O3

production limited by VOC emission.

Response: We replied to this comment in the following two points.

(1) It is correct that O3 production was VOC-limited and pNO3- production was sensitive to VOC variations. We apologize for the misstatement in the original manuscript. We had intended to state that VOC reduction is effective in mitigating both pNO3- and O3, but its effectiveness in reducing pNO3- is relatively limited when compared to NO*x* reduction. We have rewritten the relevant sentence in the revised manuscript (line 373-375) as follows:

"As mentioned above, while VOCs reduction proved effective in mitigating both  $pNO_3^-$  and  $O_3$ , its effectiveness in reducing  $pNO_3^-$  remained limited compared to NOx reduction. However, the effectiveness of NOx reduction exhibited significant regional and temporal variations."

(2) We apologize for the misunderstanding. This study primarily focuses on nitrate formation and control. While evaluating the effectiveness of NOx reduction on pNO3-, we comprehensively assessed its impact on O3 to help develop more optimized control strategies. Therefore, the NO2 limitation discussed here specifically applies to N2O5 formation that further contributing pNO3- formation, not to O3 formation. Our intention was to highlight that in the NO2-limited regime, N2O5 uptake can act as the dominant pathway for pNO3- production. The results show that daytime NOx control has a limited effect on reducing pNO3- formation and may lead to an increase in O3 concentrations. In contrast, nighttime NOx control can effectively suppress pNO3- production while avoiding O3 enhancement. To avoid misunderstanding, we have revised the abstract

(line 19-21) to clarify that the NO2 limitation refers specifically to pNO3- formation.

The modifications in the abstract are as follows:

"However, the relative contributions of  $pNO_3$  formation pathways in urban areas remain poorly quantified, particularly under the  $NO_2$ -limited regime that governs its formation (as defined by the  $NO_2/O_3$  ratio), which hinders effective particulate pollution control."

**References:**

Hu, H., Wang, H., Lu, K., Wang, J., Zheng, Z., Xu, X., Zhai, T., Chen, X., Lu, X., Fu, W., Li, X., Zeng, L., Hu, M., Zhang, Y., and Fan, S.: Variation and trend of nitrate radical reactivity towards volatile organic compounds in Beijing, China, Atmos. Chem. Phys., 23, 8211-8223, <a href="https://doi.org/10.5194/acp-23-8211-2023">https://doi.org/10.5194/acp-23-8211-2023</a>, 2023.

Thaler, R. D., Mielke, L. H., and Osthoff, H. D.: Quantification of Nitryl Chloride at Part Per Trillion Mixing Ratios by Thermal Dissociation Cavity Ring-Down Spectroscopy, Analytical Chemistry. 83, 2761-2766, <a href="https://doi.org/10.1021/ac200055z">https://doi.org/10.1021/ac200055z</a>, 2011.

Wang, H. C., Peng, C., Wang, X., Lou, S. R., Lu, K. D., Gan, G. C., Jia, X. H., Chen, X. R., Chen, J., Wang, H. L., Fan, S. J., Wang, X. M., and Tang, M. J.: N2O5 uptake onto saline mineral dust: a potential missing source of tropospheric ClNO2 in inland China, Atmospheric Chemistry and Physics. 22, 1845-1859, <a href="https://doi.org/10.5194/acp-22-1845-2022">https://doi.org/10.5194/acp-22-1845-2022</a>, 2022a.

Wang, H. C., Yuan, B., Zheng, E., Zhang, X. X., Wang, J., Lu, K. D., Ye, C. S., Yang, L., Huang, S., Hu, W. W., Yang, S. X., Peng, Y. W., Qi, J. P., Wang, S. H., He, X. J., Chen, Y. B., Li, T. G., Wang, W. J., Huangfu, Y. B., Li, X. B., Cai, M. F., Wang, X. M., and Shao, M.: Formation and impacts of nitryl chloride in Pearl River Delta, Atmospheric Chemistry and Physics. 22, 14837-14858, <a href="https://doi.org/10.5194/acp-22-14837-2022">https://doi.org/10.5194/acp-22-14837-2022</a>, 2022b.

---

## Author Comment (AC2)

Reply to comments on "Measurement report: High contribution of  $N_2O_5$  uptake to particulate nitrate formation in  $NO_2$ -limited urban areas" by Lin et al.

We would like to thank the editor and reviewers for their efforts in handling, reading, and critically reviewing our manuscript, which have helped us to further improve our manuscript. The comments on our paper are carefully addressed.

**General Comments:**

Lin et al. present an analysis of the controlling factors for particulate nitrate (pNO3-) production in Xiamen, Southeast China. Xiamen is notable compared to many other Chinese urban areas because N2O5 production there is NO2 limited, in contrast to the O3 limited conditions of other regions such as Beijing. They show that under these NO2 limited conditions, N2O5 heterogeneous uptake contributes significantly to pNO3-. These findings are significant as the conditions in the study region may be increasingly relevant to other urban areas in China, especially as emissions controls continue to change NO3, and VOC loadings. Relatedly optimal emissions control strategies to reduce pNO3- and O3 can be in conflict as elucidated in box model sensitivity simulations. Overall, this work provides useful new insights into pNO3- in the NO2 limited regime for N2O5 production. The analysis is of a high quality, and conclusions are well supported. I believe this work will be a useful addition to the literature and will likely be well suited for publication in ACP following revision and response to the comments below.

Response: We are grateful for your thoughtful comments on the manuscript and we

have made revisions accordingly. Our point-to-point responses to each comment are as follows (reviewer's comments are in black font, our responses are in blue font and our revisions in the manuscript are italic font).

**Main Comments:**

1. Was aerosol surface area density measured? If so, I would encourage the authors to also present values for the  $N_2O_5$  heterogeneous uptake coefficient ( $\gamma N_2O_5$ ) derived from the iterative box model.  $\gamma N_2O_5$  is known to depend on  $pNO_3^-$  concentrations and it could be quite interesting to see if that feedback impacts overall  $pNO_3^-$  formation from  $N_2O_5$   $\gamma N_2O_5$  values would also help with interpretation of the analytical results and iterative model skill (e.g. why is  $kN_2O_5$  so much higher in this work than in other urban areas as noted in Line 298, is this due to differences in surface area or  $\gamma N_2O_5$ )

Response: Thanks for your suggestion. Yes, we monitored the aerosol surface area (SA) concentrations in the size range of 7-300 nm under dry conditions. Since we did not apply hygroscopicity parameters to correct the data, the reported SA concentration was underestimated. For the valid observation, the average SA concentration was 110  $\mu$ m2/cm3, corresponding to a nighttime average N2O5 uptake coefficient ( $\gamma$ N2O5) of 0.223. The  $\gamma$ N2O5 should be considered as an upper limit, and the actual  $\gamma$ N2O5 could be lower. Compared with other Chinese sites ( $10^{-2} - 10^{-1}$ ), this  $\gamma$ N2O5 was relatively high (Li et al., 2025). Thus, the high uptake rate of N2O5 could be attributed to the elevated  $\gamma$ N2O5. Since our findings indicate that kN2O5 has a relatively limited impact on pNO3- formation compared to PNO3, we propose conducting further investigation

into  $\gamma N_2 O_5$  and its feedback with pNO3- through targeted case studies in future work.

2. Some additional details on the VOC measurements and the fraction of NO3 reactivity captured by the measured VOCs would be useful in the main text. Isoprene, styrene, and 2-butene have were shown to dominate VOC nitrate reactivity during winter in Beijing (Hu et al. 2023). Were those same species found to dominate NO3 reactivity here, and are any unmeasured VOC expected to matter for NO3 reactivity. More generally how do the specific VOC measured impact the discussion of pNO3- response to NOx and VOCs.

Response: We replied to this comment in the following two aspects.

(1) Thanks for your suggestion, we have provided additional details about the effect of VOCs on NO3 reactivity in the revised Section 3.3. Based on our observed VOCs (**Table S5**), the NO3 reactivity (kNO3) was calculated. The contribution of the observed VOC species to the NO3 reactivity are presented in **Figure R1** (**Figure S10** in the revised supplementary materials). Similar to previous observation in Beijing (Hu et al., 2023), the styrene, 2-butene, and isoprene were the dominant VOC species contributing to kNO3. In this work, we calculated the loss of N2O5, as shown in **eq 4** in the supplementary material, the kNO3/Keq[NO2] corresponds to the indirect chemical loss of N2O5 through NO3 chemistry. The reaction rate of kNO3/Keq[NO2] was calculated to be 0.000136 s-1, which is much smaller than that of the kN2O5 (0.00764 s-1). This indicates that the influence of VOCs on pNO3-1 formation via N2O5 uptake through the consumption of its precursors NO3 is minor, which supported the SHAP

analysis. For monoterpene species that are highly reactive with NO3 radicals, no relevant data were available in our study to access their impact on kNO3. This limitation likely led to an underestimation of the calculated kNO3, as we have highlighted the underestimation in the supplementary material (line 98-104).

The supplements of NO3 reactivity analysis in the main text (line 319-324) are as follows:

"The total concentrations of the observed VOCs (TVOCs) showed a weak negative correlation with  $N_2O_5$  uptake (**Figure 4e**). Similar to existing research (Hu et al., 2023), specific VOC species, such as styrene, 2-butene, and isoprene, can readily consume  $NO_3$  radicals (**Figure S10**), thereby inhibiting  $N_2O_5$  formation. However, the loss of  $N_2O_5$  through the reaction between VOCs and  $NO_3$  was relatively limited compared to its direct uptake, as determined by our calculations (Text S4), which supported the SHAP analysis."

(2) The response of pNO3- formation to VOCs reduction was considerably weaker than to NOx variations. Therefore, we did not focus on the detailed effects of individual anthropogenic VOC species on nitrate production. Although unmeasured monoterpene may influence NO3 reactivity and consequently pNO3- formation, these compounds are mainly emitted from biogenic sources, which are difficult to regulate through anthropogenic control. Thus, this aspect was also not discussed in detail in our study. In future work, we will select the periods with substantial indirect loss of N2O5 by NO3 and perform a more detailed analysis of the impact of specific VOC species on pNO3- formation.

Figure R1. Contribution of observed VOCs to the total NO3 reactivity (kNO3).

**Minor Comments:**

1. L19 and 29: The meaning of NO2-limited in the abstract may not be clear to the reader as these regimes have not yet been introduced or defined.

Response: Thanks for your comment. We have revised the abstract to more clearly indicate that the meaning of NO2-limited. The modifications in the revised manuscript (line 19-21) are as follows:

"However, the relative contributions of  $pNO_3^-$  formation pathways in urban areas remain poorly quantified, particularly under the  $NO_2$ -limited regime that governs its formation (as defined by the  $NO_2/O_3$  ratio), which hinders effective particulate pollution control."

2. L63: The meaning of this sentence isn't clear. Are you saying that when N2O5 dominates pNO3-, N2O5 production is typically NO2 limited or aerosol surface area is large.

Response: Yes, this is exactly what we intended to express. For clearer expression, we have revised the manuscript (line 63-65) as follows:

"However, the  $N_2O_5$  uptake served as the dominant pathway for  $pNO_3^-$  formation, typically under  $NO_2$ -limited conditions (e.g., reduced emissions during the pandemic) or under large aerosol surface areas (e.g., severe particulate pollution episodes)."

3. L78 and elsewhere: I would encourage making sure the terminology distinguishing various effects is clear throughout the manuscript. I understand the that the intended meaning is that VOC reduction will decrease the removal of NO3 by VOCs, leading to higher N2O5 production rates and therefore more pNO3- production from N2O5 heterogeneous reactions. However, the phrasing "enhancing N2O5 uptake" implies to me an increase in the first order N2O5 heterogeneous rate (kN2O5) which is independent of VOC. (also lines 279, 281)

Response: Thanks for your suggestion. We have modified the corresponding phrasing in the revised manuscript (e.g., **line 77-79** and **line 294-296**) to clearly distinguish between "N2O5 uptake processes" and "pNO3- production via N2O5 uptake".

The modifications in the main text are as follows:

"A recent study has revealed that under  $O_3$ -limited conditions for  $N_2O_5$  formation (Zhang et al., 2023), reducing NOx emissions had negligible effects, while reducing

VOCs decreased the consumption of  $NO_3$  by VOCs, thereby enhancing  $pNO_3$  formation from  $N_2O_5$  uptake." (line 77-79)

"The steep slope of the positive correlation between  $P(NO_3)$  and SHAP values indicated that  $P(NO_3)$  strongly enhances  $pNO_3^-$  formation via  $N_2O_5$  uptake." (line 294-296)

4. Line 131: while the R2 is good the slopes seem like they are far from 1. Please give values for these slopes and discuss implications.

Response: Thanks for your comment. The mean slopes of observed versus simulated  $N_2O_5$  and  $ClNO_2$  were 0.50 and 0.64, respectively, indicating that both  $N_2O_5$  and  $ClNO_2$  were underestimated in the simulations. This underestimation was mainly attributed to the model configuration in the multiphase chemical box model.

First, in the multiphase chemical box model, both dilution and dry deposition processes were included and constrained by the boundary layer height ( $k_{dilution} = k_{dilution,base} \times \frac{BLH_i}{BLH_{max}}$ ,  $k_{deposition} = \frac{k_{deposition,base}}{BLH_i}$ ). The rates of dilution and dry deposition may be overestimated. During the nighttime, when the boundary layer height is lower, the dry deposition rate becomes larger while the diffusion rate decreases, leading to lower simulated N2O5. In addition, transport process could contribute to ambient N2O5 levels. The absence of transport part in the box model may also contribute to the underestimation of simulated N2O5.

Second, a 3-day spin up was set before each model simulation to allow intermediate species to reach a stable concentration. Consequently, in addition to the

observed VOCs, some secondary chemical species formed from these VOCs were present in the model. These species could also react with NO3, thereby reducing the precursors of N2O5 and contributing to the underestimation of N2O5 in the simulation.

The underestimation of  $N_2O_5$  also led to an underestimation of ClNO2. Correspondingly, the pNO3- production via  $N_2O_5$  uptake would be underestimated. In the revised manuscript (line 142-147), we have addressed the underestimation caused by the model simulation and discussed its implications for the estimated pNO3- production via  $N_2O_5$  uptake.

The modifications in the main text are as follows:

"As shown in **Figure S3**, the model performed well in simulating the trends of  $N_2O_5$  and  $ClNO_2$  with  $R^2$  of 0.88 and 0.49, respectively. However, a systematic underestimation existed in the simulated  $N_2O_5$  and  $ClNO_2$  concentrations, which likely resulted from the model configuration including overestimated physical removal rates, elevated concentration of intermediate VOC species, or uncertainties in transport processes. Consequently, the simulated  $pNO_3$  formation from  $N_2O_5$  uptake in this study could be regarded as a lower limit."

5. Line 135:  $NO_3^-$  from  $N_2O_5$  can also partition to the gas phase as  $HNO_3$ . I don't think this is an important point for this analysis, but it is not clear that this effect would lead to an overestimation of the  $OH + NO_2$  pathway.

Response: Thank you for the note. We have removed the relevant content from the main text.

6. Fig 3: Panel A. Doesn't the right y-axis show the percent contribution not the ratio? Response: Thank you for the note. The right y-axis of Fig 3. Panel A is the percentage of N2O5 uptake to nitrate formation (%). We have adjusted the **Figure 3** in the revised manuscript.

7. Supplement L56 and L65: Were N2O5 and ClNO2 calibrated through the full 2 meter stainless steel inlet used for the ambient observations? If not, was an inlet loss rate determined. N2O5 loss on that length of stainless steel could be substantial.

Response: We apologize for the incorrect description in the previous version of the supplementary material. A long perfluoroalkoxy (PFA) tube with a length of nearly 2 meters and a 1/4-inch inner diameter was used for sampling, not the 2-meter stainless steel one. In order to minimize the effect of particles deposited on the surface of the sampling inlet, the tube was cleaned by deionized water and dried by nitrogen flow once a week. In the calibration process, the standard gas was also delivered to the instrument through the PFA tube, consistent with the configuration for the field measurement. To better clarify the operation and calibration of the CIMS instrument, we have moved the relevant content from the supplementary material to the main text. The revised text (line 112-127) is provided below.

"A nearly 2-meter long perfluoroalkoxy (PFA) tube with a 1/4-inch inner diameter was used for sampling. The total sampling flow rate was set as 10 standard liters per minute (SLPM), of which only 2SLPM was diverted to the CIMS. A nitrogen  $(N_2)$  flow

permeation tube, passed through a soft X-ray source (Tofwerk AG, P-type) to generate reagent ions  $\Gamma$ . The  $\Gamma$  was combined with the target gas in an ion molecule reaction (IMR) chamber and then detected by the ToF-CIMS. Ambient  $N_2O_5$  and  $CINO_2$  were detected as the  $I(N_2O_5)^-$  and  $I(CINO_2)^-$  clusters at 235 and 208 m/z. The detailed calibration procedures of  $N_2O_5$  and  $CINO_2$  are described in **Text S2**, following established methods (Wang et al., 2022c; Wang et al., 2022b; Thaler et al., 2011). Briefly,  $N_2O_5$  was generated from the reaction between  $O_3$  and excessive  $NO_2$ , while  $CINO_2$  was synthesized via the reaction of  $Cl_2$  (6 ppm in  $N_2$ ) with a moist mixture of

NaNO2 and NaCl. The calibration curves for N2O5 and ClNO2 at different RH are

shown in **Figure S2**, with mean sensitivities of  $0.110 \pm 0.063$  and  $0.055 \pm 0.018$

ncps/ppb, respectively. The instrument background was determined by introducing dry

 $N_2$  into the inlet for 20 min. Based on three times the standard deviation (3 $\sigma$ ) of the

background signal, the typical 1-minute detection limits for  $N_2O_5$  and  $ClNO_2$  were

(99.999%, 2.7 SLPM), carrying methyl iodide (CH3I) vapor released from a heated

8. Supplement L62: IClNO2 is at m/z 208

estimated to be 1.3 and 0.61 ppt, respectively."

Response: Thank you for the note. We have corrected it.

9. Supplement L82: At what averaging time?

Response: Thank you for the comment. The averaging time is 1 minute and we have added it in the revised main text (line 125-127) as follows.

"Based on three times the standard deviation (3 $\sigma$ ) of the background signal, the typical 1-minute detection limits for  $N_2O_5$  and  $ClNO_2$  were estimated to be 1.3 and 0.61 ppt, respectively."

10. Figure S2: These sensitivities are notably quite low compared to typical Iodide CIMS instruments. Also, the LODs quoted in line L82 seem very good given the poor sensitivity. Can you expand further on how these values were derived.

Response: We replied to this comment in the following two aspects.

(1) In **Figure S2**, the sensitivities appear lower due to the normalization of  $N_2O_5$  and  $CINO_2$  signals applied in the calibration curves. The normalized signals of  $N_2O_5$  and  $CINO_2$  are calculated as  $N_2O_5(ncps) = \frac{(IN_2O_5)^5}{\Gamma + (IH_2O)^5}$  and  $CINO_2(ncps) = \frac{(CINO_2)^5}{\Gamma + (IH_2O)^5}$ , respectively. The signal intensity of  $[I^5 + (IH_2O)^5]$  was approximately on the order of  $10^5$  counts. Consequently, compared with the signal  $(IN_2O_5)^5$ , the normalized signals were quite low. Thus, the sensitivities appeared relatively low. In our work, the mean sensitivities of  $N_2O_5$  and  $CINO_2$  were  $0.110 \pm 0.063$  and  $0.055 \pm 0.018$  ncps/ppb, respectively, which are comparable to those reported in existing research (see in Figure R2). To avoid misunderstanding, we have revised the description of Figure S2 (line 172-174) in the revised supplementary material) to emphasize that the signals represent normalized results, and we have presented the corresponding sensitivities in the main text (line 122-124).

The modifications are presented below.

"In panels (a) and (b), the signals of  $N_2O_5$  and  $ClNO_2$  are normalized signals,

which were calculated according equation  $N_2O_5(ncps) = \frac{(IN_2O_5)^-}{\Gamma + (IH_2O)^-}$  and equation  $CINO_2(ncps) = \frac{(CINO_2)^-}{\Gamma + (IH_2O)^-}$ , respectively." (line 172-174 in the revised supplementary material)

"The final calibration curves for  $N_2O_5$  and  $ClNO_2$  at different RH are shown as **Figure S2** with mean sensitivities of  $0.110 \pm 0.063$  and  $0.055 \pm 0.018$  ncps/ppb, respectively." (line 122-124)

**Figure R2.** CIMS sensitivities as a function of RH for N2O5 and ClNO2 reported in the existing study by Wang et al. (Wang et al., 2022a).

(2) As for the LOD, it was calculated based on the standard deviation of the background signal and the sensitivity. The background signals of the CIMS instrument were determined by introducing dry  $N_2$  into the inlet for a duration of 20 min. According

to three times the standard deviation  $(3\sigma)$  of the background signal, the typical detection limit of  $N_2O_5$  and  $ClNO_2$  for 1 min were estimated. In the revised main text, we have added the details (line 125-127) as follows:

"The background signals of the CIMS instrument ascertained by introducing dry  $N_2$  into the inlet for a duration of 20 min. According to three times the standard deviation (3 $\sigma$ ) of the background signal, the typical detection limit of  $N_2O_5$  and  $CINO_2$  for 1 min were estimated to be 1.3 and 0.61 ppt, respectively."

11. Supplement L85: What time resolution data was used for the iterative box model Response: Thank you for the comment. The time resolution of the input data for the iterative box model is one hour. We have added this detail in the revised supplementary material (line 82-83).

The modifications are as below.

"Notably, the input data for the iterative box model have a time resolution of 1 hour."

**References:**

Hu, H., Wang, H., Lu, K., Wang, J., Zheng, Z., Xu, X., Zhai, T., Chen, X., Lu, X., Fu, W., Li, X., Zeng, L., Hu, M., Zhang, Y., and Fan, S.: Variation and trend of nitrate radical reactivity towards volatile organic compounds in Beijing, China, Atmos. Chem. Phys., 23, 8211-8223, <a href="https://doi.org/10.5194/acp-23-8211-2023">https://doi.org/10.5194/acp-23-8211-2023</a>, 2023.

Li, J., Zhai, T., Chen, X., Wang, H., Xie, S., Chen, S., Li, C., Gong, Y., Dong, H., and Lu, K.: Direct measurement of  $N_2O_5$  heterogeneous uptake coefficients on atmospheric aerosols in southwestern China and evaluation of current parameterizations, Atmos. Chem. Phys., 25, 6395-6406, <a href="https://doi.org/10.5194/acp-25-6395-2025">https://doi.org/10.5194/acp-25-6395-2025</a>, 2025.

Thaler, R. D., Mielke, L. H., and Osthoff, H. D.: Quantification of Nitryl Chloride at Part Per Trillion Mixing Ratios by Thermal Dissociation Cavity Ring-Down Spectroscopy, Analytical Chemistry. 83, 2761-2766, <a href="https://doi.org/10.1021/ac200055z">https://doi.org/10.1021/ac200055z</a>, 2011.

Wang, H., Peng, C., Wang, X., Lou, S., Lu, K., Gan, G., Jia, X., Chen, X., Chen, J., Wang, H., Fan, S., Wang, X., and Tang, M.: N2O5 uptake onto saline mineral dust: a potential missing source of tropospheric ClNO2 in inland China, Atmos. Chem. Phys., 22, 1845-1859, <a href="https://doi.org/10.5194/acp-22-1845-2022">https://doi.org/10.5194/acp-22-1845-2022</a>, 2022a.

Wang, H. C., Peng, C., Wang, X., Lou, S. R., Lu, K. D., Gan, G. C., Jia, X. H., Chen, X. R., Chen, J., Wang, H. L., Fan, S. J., Wang, X. M., and Tang, M. J.: N2O5 uptake onto saline mineral dust: a potential missing source of tropospheric ClNO2 in inland China, Atmospheric Chemistry and Physics. 22, 1845-1859, <a href="https://doi.org/10.5194/acp-22-1845-2022">https://doi.org/10.5194/acp-22-1845-2022</a>, 2022b.

Wang, H. C., Yuan, B., Zheng, E., Zhang, X. X., Wang, J., Lu, K. D., Ye, C. S., Yang, L., Huang, S., Hu, W. W., Yang, S. X., Peng, Y. W., Qi, J. P., Wang, S. H., He, X. J., Chen, Y. B., Li, T. G., Wang, W. J., Huangfu, Y. B., Li, X. B., Cai, M. F., Wang, X. M., and Shao, M.: Formation and impacts of nitryl chloride in Pearl River Delta, Atmospheric Chemistry and Physics. 22, 14837-14858, <a href="https://doi.org/10.5194/acp-22-14837-2022">https://doi.org/10.5194/acp-22-14837-2022</a>, 2022c.

Zhang, Y. N., Wang, H. L., Huang, L. B., Qiao, L. P., Zhou, M., Mu, J. S., Wu, C., Zhu, Y. J., Shen, H. Q., Huang, C., Wang, G. H., Wang, T., Wang, W. X., and Xue, L. K.: Double-Edged Role of VOCs Reduction in Nitrate Formation: Insights from Observations during the China International Import Expo 2018, Environmental Science & Technology. 57, 15979-15989, <a href="https://doi.org/10.1021/acs.est.3c04629">https://doi.org/10.1021/acs.est.3c04629</a>, 2023.

---

## Author Response (AR1)

**Reply to comments on "Measurement report: High contribution of N$_2$O$_5$ uptake to particulate nitrate formation in NO$_2$-limited urban areas" by Lin et al.**

We would like to thank the editor and reviewers for their efforts in handling, reading, and critically reviewing our manuscript, which have helped us to further improve our manuscript. The comments on our paper are carefully addressed.

**Response to Reviewer #1:**

**General Comments:**

This study investigates the features of pNO$_3^-$ production at a typical site in southeastern China. The nighttime N$_2$O$_5$ uptake was found to be efficient enough and play an increasing role in pNO$_3^-$ formation. Multiple methods are applied to demonstrating the dominant effect of PNO$_3$ or precursor concentrations, in N$_2$O$_5$ uptake process. The manuscript is well organized. I recommend acceptance after carefully addressing the following concerns when revising this manuscript.

Response: We are grateful for your thoughtful comments on the manuscript and we have made careful revisions accordingly. Our point-to-point responses to each comment are as follows (reviewer's comments are in black font, our responses are in blue font and our revisions in the manuscript are italic font).

1. Method 2.1: I suggest more description of instrument deployment at least for I-Tof-CIMS in main text. It would be better to briefly introduce the sampling setup, calibration frequency and the variation of calibration factor, such that the

reasonableness of measurements could be readily assessed.

Response: Thank you for this suggestion. We have added the detailed operation and calibration procedure for the I-TOF-CIMS in the revised manuscript (**line 112-127**), as follows:

"*A nearly 2-meter long perfluoroalkoxy (PFA) tube with a 1/4-inch inner diameter was used for sampling. The total sampling flow rate was set as 10 standard liters per minute (SLPM), of which only 2SLPM was diverted to the CIMS. A nitrogen ($N_2$) flow (99.999%, 2.7 SLPM), carrying methyl iodide ($CH_3I$) vapor released from a heated permeation tube, passed through a soft X-ray source (Tofwerk AG, P-type) to generate reagent ions $I^-$. The $I^-$ was combined with the target gas in an ion molecule reaction (IMR) chamber and then detected by the ToF-CIMS. Ambient $N_2O_5$ and $ClNO_2$ were detected as the $I(N_2O_5)^-$ and $I(ClNO_2)^-$ clusters at 235 and 208 m/z. The detailed calibration procedures of $N_2O_5$ and $ClNO_2$ are described in **Text S2**, following established methods (Wang et al., 2022c; Wang et al., 2022b; Thaler et al., 2011). Briefly, $N_2O_5$ was generated from the reaction between $O_3$ and excessive $NO_2$, while $ClNO_2$ was synthesized via the reaction of $Cl_2$ (6 ppm in $N_2$) with a moist mixture of $NaNO_2$ and NaCl. The calibration curves for $N_2O_5$ and $ClNO_2$ at different RH are shown in **Figure S2,** with mean sensitivities of 0.110 ± 0.063 and 0.055 ± 0.018 ncps/ppb, respectively. The instrument background was determined by introducing dry $N_2$ into the inlet for 20 min. Based on three times the standard deviation ($3\sigma$) of the background signal, the typical 1-minute detection limits for $N_2O_5$ and $ClNO_2$ were estimated to be 1.3 and 0.61 ppt, respectively.*"

2. Method 2.2: The algorithm proposed by Wagner 2013 should be termed as iterative box model instead of interactive box model.

Response: Thank you for the note. We have corrected it in the revised manuscript (**line 130,132 and 139**).

3. Line 212: Suggest clearly stating what simulate parameters exhibited good agreement here.

Response: As you suggested, we have rewritten the sentence in the revised manuscript (**line 225-228**) as follows:

*"As shown in **Figure S7,** the simulated $kN_2O_5$ and $\varphi ClNO_2$ exhibited good agreement with the classical steady-state method ($R^2$ = 0.76 and 0.73, respectively), demonstrating the model's capability to characterize heterogeneous uptake processes and thereby effectively evaluate $pNO_3^-$ formation processes."*

4. Line 218-230: A direct comparison of $pNO_3^-$ formation rate is also helpful to indicate the characteristics of $pNO_3^-$ formation at this site.

Response: Thank you for the suggestion. We have added a direct comparison of $pNO_3^-$ formation rate in the revised manuscript (**line 234-238**) as follows:

*"To exclude year-specific effects, we further analyzed $pNO_3^-$ formation during the winters from 2019 to 2023. The results revealed that the $pNO_3^-$ formation rates via $N_2O_5$ uptake (0.75 – 1.40 $\mu g\ m^{-3}\ h^{-1}$) were comparable to those from the OH + $NO_2$ reaction*

*(0.88 – 1.66 μg m$^{-3}$ h$^{-1}$; **Figure 3a**), with the N$_2$O$_5$ uptake pathway consistently accounting for approximately half of the total pNO$_3^-$ formation in the study area (**Figure 3b**)."*

5. Line 241: Please revise the text font of citation.

Response: Thank you for the note. We have corrected it in the revised manuscript (**line 253**).

6. Line 307-314: The SHAP of TVOC exhibit minor impact on N$_2$O$_5$ uptake and correlate not so well with its concentration. Replacing TVOCs with specific VOC species, such as monoterpene and styrene, could provide better correlation of this feature.

Response: Thank you for the comment. We agree that specific VOCs species influence N$_2$O$_5$ formation due to the higher reactivity of NO$_3$ toward them. In this work, the loss of N$_2$O$_5$ was calculated using **eqs. R1-R2** (**eqs. S3-S4** in the supplementary material), where kN$_2$O$_5$ represents the rate of N$_2$O$_5$ uptake, and kNO$_3$/Keq[NO$_2$] corresponds to the indirect chemical loss of N$_2$O$_5$ through NO$_3$ chemistry. As shown in **Table S6**, the reaction rate of kNO$_3$/Keq[NO$_2$] was calculated to be 0.000136 s$^{-1}$, which is much smaller than that of kN$_2$O$_5$ (0.00764 s$^{-1}$). This indicates that the loss of N$_2$O$_5$ through the consumption of its precursors NO$_3$ by VOCs is relatively limited compared to its direct uptake. Considering this finding and the risk of model overfitting when including too many variables, we used TVOCs as a simplified indicator to represent the effect of

VOCs on pNO$_3^-$ formation via N$_2$O$_5$ uptake. As a result, the low SHAP value of TVOCs is consistent with their limited influence on N$_2$O$_5$ removal, as determined by our calculations. We have included a detailed discussion in the supplementary material (**line 98-104**) and provided corresponding explanations in the revised manuscript (**line 319-324**).

$$\tau_{N_2O_5} = \frac{[N_2O_5]}{K_1(T)[NO_2][O_3]} \tag{R1}$$

$$(\tau_{N_2O_5})^{-1} = kN_2O_5 + \frac{k_{NO_3}}{K_{eq}[NO_2]} \tag{R2}$$

The corresponding explanations in the main text are as follows:

*"The total concentrations of the observed VOCs (TVOCs) showed a weak negative correlation with N$_2$O$_5$ uptake (**Figure 4e**). Similar to existing research (Hu et al., 2023), specific VOC species, such as styrene, 2-butene, and isoprene, can readily consume NO$_3$ radicals (**Figure S10**), thereby inhibiting N$_2$O$_5$ formation. However, the loss of N$_2$O$_5$ through the reaction between VOCs and NO$_3$ was relatively limited compared to its direct uptake, as determined by our calculations (Text S4), which supported the SHAP analysis."*

7. Section 3.4: The response of pNO$_3^-$ and O$_3$ production rate to precursors is well investigated, while I figure out two confusing points in discussion part. First, the production of O$_3$ was clearly proved to be VOC-limited, resulting in effective mitigation on O$_3$ by reducing VOC. Meanwhile, the pNO$_3^-$ production also shows larger sensitivity to VOC variation. However, the authors claim that the effect of VOC reduction is limited in mitigating both pNO$_3^-$ and O$_3$, which is confusing. Second, the

title of this manuscript, a $NO_2$-limited region, seems contradictory to the finding of $O_3$ production limited by VOC emission.

Response: We replied to this comment in the following two points.

(1) It is correct that $O_3$ production was VOC-limited and $pNO_3^-$ production was sensitive to VOC variations. We apologize for the misstatement in the original manuscript. We had intended to state that VOC reduction is effective in mitigating both $pNO_3^-$ and $O_3$, but its effectiveness in reducing $pNO_3^-$ is relatively limited when compared to NO$x$ reduction. We have rewritten the relevant sentence in the revised manuscript (**line 373-375**) as follows:

*"As mentioned above, while VOCs reduction proved effective in mitigating both $pNO_3^-$ and $O_3$, its effectiveness in reducing $pNO_3^-$ remained limited compared to NOx reduction. However, the effectiveness of NOx reduction exhibited significant regional and temporal variations."*

(2) We apologize for the misunderstanding. This study primarily focuses on nitrate formation and control. While evaluating the effectiveness of NO$x$ reduction on $pNO_3^-$, we comprehensively assessed its impact on $O_3$ to help develop more optimized control strategies. Therefore, the $NO_2$ limitation discussed here specifically applies to $N_2O_5$ formation that further contributing $pNO_3^-$ formation, not to $O_3$ formation. Our intention was to highlight that in the $NO_2$-limited regime, $N_2O_5$ uptake can act as the dominant pathway for $pNO_3^-$ production. The results show that daytime NO$x$ control has a limited effect on reducing $pNO_3^-$ formation and may lead to an increase in $O_3$ concentrations. In contrast, nighttime NO$x$ control can effectively suppress $pNO_3^-$ production while

avoiding $O_3$ enhancement. To avoid misunderstanding, we have revised the abstract (**line 19-21**) to clarify that the $NO_2$ limitation refers specifically to $pNO_3^-$ formation. The modifications in the abstract are as follows:

*"However, the relative contributions of $pNO_3^-$ formation pathways in urban areas remain poorly quantified, particularly under the $NO_2$-limited regime that governs its formation (as defined by the $NO_2/O_3$ ratio), which hinders effective particulate pollution control."*

**Response to Reviewer #2:**

**General Comments:**

Lin et al. present an analysis of the controlling factors for particulate nitrate ($pNO_3^-$) production in Xiamen, Southeast China. Xiamen is notable compared to many other Chinese urban areas because $N_2O_5$ production there is $NO_2$ limited, in contrast to the $O_3$ limited conditions of other regions such as Beijing. They show that under these $NO_2$ limited conditions, $N_2O_5$ heterogeneous uptake contributes significantly to $pNO_3^-$. These findings are significant as the conditions in the study region may be increasingly relevant to other urban areas in China, especially as emissions controls continue to change $NOx$, $O_3$, and VOC loadings. Relatedly optimal emissions control strategies to reduce $pNO_3-$ and $O_3$ can be in conflict as elucidated in box model sensitivity simulations. Overall, this work provides useful new insights into $pNO_3-$ in the $NO_2$ limited regime for $N_2O_5$ production. The analysis is of a high quality, and conclusions are well supported. I believe this work will be a useful addition to the literature and will likely be well suited for publication in ACP following revision and response to the comments below.

Response: We are grateful for your thoughtful comments on the manuscript and we have made revisions accordingly. Our point-to-point responses to each comment are as follows (reviewer's comments are in black font, our responses are in blue font and our revisions in the manuscript are italic font).

**Main Comments:**

1. Was aerosol surface area density measured? If so, I would encourage the authors to also present values for the $N_2O_5$ heterogeneous uptake coefficient ($\gamma N_2O_5$) derived from the iterative box model. $\gamma N_2O_5$ is known to depend on $pNO_3^-$ concentrations and it could be quite interesting to see if that feedback impacts overall $pNO_3^-$ formation from $N_2O_5$ $\gamma N_2O_5$ values would also help with interpretation of the analytical results and iterative model skill (e.g. why is $kN_2O_5$ so much higher in this work than in other urban areas as noted in Line 298, is this due to differences in surface area or $\gamma N_2O_5$)

Response: Thanks for your suggestion. Yes, we monitored the aerosol surface area (SA) concentrations in the size range of 7-300 nm under dry conditions. Since we did not apply hygroscopicity parameters to correct the data, the reported SA concentration was underestimated. For the valid observation, the average SA concentration was 110 $\mu m^2/cm^3$, corresponding to a nighttime average $N_2O_5$ uptake coefficient ($\gamma N_2O_5$) of 0.223. The $\gamma N_2O_5$ should be considered as an upper limit, and the actual $\gamma N_2O_5$ could be lower. Compared with other Chinese sites ($10^{-2} - 10^{-1}$), this $\gamma N_2O_5$ was relatively high (Li et al., 2025). Thus, the high uptake rate of $N_2O_5$ could be attributed to the elevated $\gamma N_2O_5$. Since our findings indicate that $kN_2O_5$ has a relatively limited impact on $pNO_3^-$ formation compared to $PNO_3$, we propose conducting further investigation into $\gamma N_2O_5$ and its feedback with $pNO_3^-$ through targeted case studies in future work.

2. Some additional details on the VOC measurements and the fraction of $NO_3$ reactivity captured by the measured VOCs would be useful in the main text. Isoprene, styrene, and 2-butene have were shown to dominate VOC nitrate reactivity during winter in

Beijing (Hu et al. 2023). Were those same species found to dominate $NO_3$ reactivity here, and are any unmeasured VOC expected to matter for $NO_3$ reactivity. More generally how do the specific VOC measured impact the discussion of $pNO_3^-$ response to $NOx$ and VOCs.

Response: We replied to this comment in the following two aspects.

(1) Thanks for your suggestion, we have provided additional details about the effect of VOCs on $NO_3$ reactivity in the revised Section 3.3. Based on our observed VOCs (**Table S5**), the $NO_3$ reactivity ($kNO_3$) was calculated. The contribution of the observed VOC species to the $NO_3$ reactivity are presented in **Figure R1 (Figure S10 in the revised supplementary materials)**. Similar to previous observation in Beijing (Hu et al., 2023), the styrene, 2-butene, and isoprene were the dominant VOC species contributing to $kNO_3$. In this work, we calculated the loss of $N_2O_5$, as shown in **eq 4** in the supplementary material, the $kNO_3/Keq[NO_2]$ corresponds to the indirect chemical loss of $N_2O_5$ through $NO_3$ chemistry. The reaction rate of $kNO_3/Keq[NO_2]$ was calculated to be 0.000136 $s^{-1}$, which is much smaller than that of the $kN_2O_5$ (0.00764 $s^{-1}$). This indicates that the influence of VOCs on $pNO_3^-$ formation via $N_2O_5$ uptake through the consumption of its precursors $NO_3$ is minor, which supported the SHAP analysis. For monoterpene species that are highly reactive with $NO_3$ radicals, no relevant data were available in our study to access their impact on $kNO_3$. This limitation likely led to an underestimation of the calculated $kNO_3$, as we have highlighted the underestimation in the supplementary material (**line 98-104**).

The supplements of $NO_3$ reactivity analysis in the main text (**line 319-324**) are as

follows:

*"The total concentrations of the observed VOCs (TVOCs) showed a weak negative correlation with $N_2O_5$ uptake (**Figure 4e**). Similar to existing research (Hu et al., 2023), specific VOC species, such as styrene, 2-butene, and isoprene, can readily consume $NO_3$ radicals (**Figure S10**), thereby inhibiting $N_2O_5$ formation. However, the loss of $N_2O_5$ through the reaction between VOCs and $NO_3$ was relatively limited compared to its direct uptake, as determined by our calculations (Text S4), which supported the SHAP analysis."*

(2) The response of $pNO_3^-$ formation to VOCs reduction was considerably weaker than to $NOx$ variations. Therefore, we did not focus on the detailed effects of individual anthropogenic VOC species on nitrate production. Although unmeasured monoterpene may influence $NO_3$ reactivity and consequently $pNO_3^-$ formation, these compounds are mainly emitted from biogenic sources, which are difficult to regulate through anthropogenic control. Thus, this aspect was also not discussed in detail in our study. In future work, we will select the periods with substantial indirect loss of $N_2O_5$ by $NO_3$ and perform a more detailed analysis of the impact of specific VOC species on $pNO_3^-$ formation.

[Figure]

**Figure R1.** Contribution of observed VOCs to the total $NO_3$ reactivity ($kNO_3$).

**Minor Comments:**

1. L19 and 29: The meaning of $NO_2$-limited in the abstract may not be clear to the reader as these regimes have not yet been introduced or defined.

Response: Thanks for your comment. We have revised the abstract to more clearly indicate that the meaning of $NO_2$-limited. The modifications in the revised manuscript **(line 19-21)** are as follows:

*"However, the relative contributions of $pNO_3^-$ formation pathways in urban areas remain poorly quantified, particularly under the $NO_2$-limited regime that governs its formation (as defined by the $NO_2/O_3$ ratio), which hinders effective particulate pollution control."*

2. L63: The meaning of this sentence isn't clear. Are you saying that when $N_2O_5$ dominates $pNO_3^-$, $N_2O_5$ production is typically $NO_2$ limited or aerosol surface area is large.

Response: Yes, this is exactly what we intended to express. For clearer expression, we have revised the manuscript (**line 63-65**) as follows:

*"However, the $N_2O_5$ uptake served as the dominant pathway for $pNO_3^-$ formation, typically under $NO_2$-limited conditions (e.g., reduced emissions during the pandemic) or under large aerosol surface areas (e.g., severe particulate pollution episodes)."*

3. L78 and elsewhere: I would encourage making sure the terminology distinguishing various effects is clear throughout the manuscript. I understand the that the intended meaning is that VOC reduction will decrease the removal of $NO_3$ by VOCs, leading to higher $N_2O_5$ production rates and therefore more $pNO_3^-$ production from $N_2O_5$ heterogeneous reactions. However, the phrasing "enhancing $N_2O_5$ uptake" implies to me an increase in the first order $N_2O_5$ heterogenous rate ($kN_2O_5$) which is independent of VOC. (also lines 279, 281)

Response: Thanks for your suggestion. We have modified the corresponding phrasing in the revised manuscript (e.g., **line 77-79** and **line 294-296**) to clearly distinguish between "$N_2O_5$ uptake processes" and "$pNO_3^-$ production via $N_2O_5$ uptake".

The modifications in the main text are as follows:

*"A recent study has revealed that under $O_3$-limited conditions for $N_2O_5$ formation (Zhang et al., 2023), reducing NOx emissions had negligible effects, while reducing*

*VOCs decreased the consumption of NO₃ by VOCs, thereby enhancing pNO₃⁻ formation*

*from N₂O₅ uptake." (line 77-79)*

*"The steep slope of the positive correlation between P(NO₃) and SHAP values*

*indicated that P(NO₃) strongly enhances pNO₃⁻ formation via N₂O₅ uptake." (line 294-*

*296)*

4. Line 131: while the R2 is good the slopes seem like they are far from 1. Please give values for these slopes and discuss implications.

Response: Thanks for your comment. The mean slopes of observed versus simulated $N_2O_5$ and $ClNO_2$ were 0.50 and 0.64, respectively, indicating that both $N_2O_5$ and $ClNO_2$ were underestimated in the simulations. This underestimation was mainly attributed to the model configuration in the multiphase chemical box model.

First, in the multiphase chemical box model, both dilution and dry deposition processes were included and constrained by the boundary layer height ($k_{dilution} = k_{dilution,base} \times \frac{BLH_i}{BLH_{max}}$, $k_{deposition} = \frac{k_{deposition,base}}{BLH_i}$). The rates of dilution and dry deposition may be overestimated. During the nighttime, when the boundary layer height is lower, the dry deposition rate becomes larger while the diffusion rate decreases, leading to lower simulated $N_2O_5$. In addition, transport process could contribute to ambient $N_2O_5$ levels. The absence of transport part in the box model may also contribute to the underestimation of simulated $N_2O_5$.

Second, a 3-day spin up was set before each model simulation to allow intermediate species to reach a stable concentration. Consequently, in addition to the

observed VOCs, some secondary chemical species formed from these VOCs were present in the model. These species could also react with $NO_3$, thereby reducing the precursors of $N_2O_5$ and contributing to the underestimation of $N_2O_5$ in the simulation.

The underestimation of $N_2O_5$ also led to an underestimation of $ClNO_2$. Correspondingly, the $pNO_3^-$ production via $N_2O_5$ uptake would be underestimated. In the revised manuscript (**line 142-147**), we have addressed the underestimation caused by the model simulation and discussed its implications for the estimated $pNO_3^-$ production via $N_2O_5$ uptake.

The modifications in the main text are as follows:

*"As shown in **Figure S3**, the model performed well in simulating the trends of $N_2O_5$ and $ClNO_2$ with $R^2$ of 0.88 and 0.49, respectively. However, a systematic underestimation existed in the simulated $N_2O_5$ and $ClNO_2$ concentrations, which likely resulted from the model configuration including overestimated physical removal rates, elevated concentration of intermediate VOC species, or uncertainties in transport processes. Consequently, the simulated $pNO_3^-$ formation from $N_2O_5$ uptake in this study could be regarded as a lower limit."*

5. Line 135: $NO_3^-$ from $N_2O_5$ can also partition to the gas phase as $HNO_3$. I don't think this is an important point for this analysis, but it is not clear that this effect would lead to an overestimation of the $OH + NO_2$ pathway.

Response: Thank you for the note. We have removed the relevant content from the main text.

6. Fig 3: Panel A. Doesn't the right y-axis show the percent contribution not the ratio?

Response: Thank you for the note. The right y-axis of Fig 3. Panel A is the percentage of $N_2O_5$ uptake to nitrate formation (%). We have adjusted the **Figure 3** in the revised manuscript.

7. Supplement L56 and L65: Were $N_2O_5$ and $ClNO_2$ calibrated through the full 2 meter stainless steel inlet used for the ambient observations? If not, was an inlet loss rate determined. $N_2O_5$ loss on that length of stainless steel could be substantial.

Response: We apologize for the incorrect description in the previous version of the supplementary material. A long perfluoroalkoxy (PFA) tube with a length of nearly 2 meters and a 1/4 -inch inner diameter was used for sampling, not the 2-meter stainless steel one. In order to minimize the effect of particles deposited on the surface of the sampling inlet, the tube was cleaned by deionized water and dried by nitrogen flow once a week. In the calibration process, the standard gas was also delivered to the instrument through the PFA tube, consistent with the configuration for the field measurement. To better clarify the operation and calibration of the CIMS instrument, we have moved the relevant content from the supplementary material to the main text. The revised text (**line 112-127**) is provided below.

*"A nearly 2-meter long perfluoroalkoxy (PFA) tube with a 1/4-inch inner diameter was used for sampling. The total sampling flow rate was set as 10 standard liters per minute (SLPM), of which only 2SLPM was diverted to the CIMS. A nitrogen ($N_2$) flow*

*(99.999%, 2.7 SLPM), carrying methyl iodide (CH₃I) vapor released from a heated permeation tube, passed through a soft X-ray source (Tofwerk AG, P-type) to generate reagent ions I⁻. The I⁻ was combined with the target gas in an ion molecule reaction (IMR) chamber and then detected by the ToF-CIMS. Ambient $N_2O_5$ and $ClNO_2$ were detected as the $I(N_2O_5)^-$ and $I(ClNO_2)^-$ clusters at 235 and 208 m/z. The detailed calibration procedures of $N_2O_5$ and $ClNO_2$ are described in **Text S2**, following established methods (Wang et al., 2022c; Wang et al., 2022b; Thaler et al., 2011). Briefly, $N_2O_5$ was generated from the reaction between $O_3$ and excessive $NO_2$, while $ClNO_2$ was synthesized via the reaction of $Cl_2$ (6 ppm in $N_2$) with a moist mixture of $NaNO_2$ and NaCl. The calibration curves for $N_2O_5$ and $ClNO_2$ at different RH are shown in **Figure S2,** with mean sensitivities of 0.110 ± 0.063 and 0.055 ± 0.018 ncps/ppb, respectively. The instrument background was determined by introducing dry $N_2$ into the inlet for 20 min. Based on three times the standard deviation (3σ) of the background signal, the typical 1-minute detection limits for $N_2O_5$ and $ClNO_2$ were estimated to be 1.3 and 0.61 ppt, respectively."*

8. Supplement L62: $IClNO_2^-$ is at m/z 208

Response: Thank you for the note. We have corrected it.

9. Supplement L82: At what averaging time?

Response: Thank you for the comment. The averaging time is 1 minute and we have added it in the revised main text (**line 125-127**) as follows.

*"Based on three times the standard deviation (3σ) of the background signal, the typical 1-minute detection limits for $N_2O_5$ and $ClNO_2$ were estimated to be 1.3 and 0.61 ppt, respectively."*

10. Figure S2: These sensitivities are notably quite low compared to typical Iodide CIMS instruments. Also, the LODs quoted in line L82 seem very good given the poor sensitivity. Can you expand further on how these values were derived.

Response: We replied to this comment in the following two aspects.

(1) In **Figure S2**, the sensitivities appear lower due to the normalization of $N_2O_5$ and $ClNO_2$ signals applied in the calibration curves. The normalized signals of $N_2O_5$ and $ClNO_2$ are calculated as $N_2O_5(ncps)=\frac{(IN_2O_5)^-}{I^-+(IH_2O)^-}$ and $ClNO_2(ncps)=\frac{(ClNO_2)^-}{I^-+(IH_2O)^-}$, respectively. The signal intensity of $[I^- + (IH_2O)^-]$ was approximately on the order of $10^5$ counts. Consequently, compared with the signal $(IN_2O_5)^-$, the normalized signals were quite low. Thus, the sensitivities appeared relatively low. In our work, the mean sensitivities of $N_2O_5$ and $ClNO_2$ were $0.110 \pm 0.063$ and $0.055 \pm 0.018$ ncps/ppb, respectively, which are comparable to those reported in existing research (**see in Figure R2**). To avoid misunderstanding, we have revised the description of **Figure S2 (line 172-174)** in the revised supplementary material) to emphasize that the signals represent normalized results, and we have presented the corresponding sensitivities in the main text (**line 122-124**).

The modifications are presented below.

*"In panels (a) and (b), the signals of $N_2O_5$ and $ClNO_2$ are normalized signals,*

which were calculated according equation $N_2O_5(ncps) = \frac{(IN_2O_5)^-}{I^- + (IH_2O)^-}$ and equation $ClNO_2(ncps) = \frac{(ClNO_2)^-}{I^- + (IH_2O)^-}$, respectively." (line 172-174 in the revised supplementary material)

"The final calibration curves for $N_2O_5$ and $ClNO_2$ at different RH are shown as **Figure S2** with mean sensitivities of 0.110 ± 0.063 and 0.055 ± 0.018 ncps/ppb, respectively." (line 122-124)

[Figure]

**Figure R2.** CIMS sensitivities as a function of RH for $N_2O_5$ and $ClNO_2$ reported in the existing study by Wang et al (Wang et al., 2022a).

(2) As for the LOD, it was calculated based on the standard deviation of the background signal and the sensitivity. The background signals of the CIMS instrument were determined by introducing dry $N_2$ into the inlet for a duration of 20 min. According

to three times the standard deviation (3σ) of the background signal, the typical detection limit of N₂O₅ and ClNO₂ for 1 min were estimated. In the revised main text, we have added the details (**line 125-127**) as follows:

*"The background signals of the CIMS instrument ascertained by introducing dry N₂ into the inlet for a duration of 20 min. According to three times the standard deviation (3σ) of the background signal, the typical detection limit of N₂O₅ and ClNO₂ for 1 min were estimated to be 1.3 and 0.61 ppt, respectively."*

11. Supplement L85: What time resolution data was used for the iterative box model

Response: Thank you for the comment. The time resolution of the input data for the iterative box model is one hour. We have added this detail in the revised supplementary material (**line 82-83**).

The modifications are as below.

*"Notably, the input data for the iterative box model have a time resolution of 1 hour."*

**Reference:**

Hu, H., Wang, H., Lu, K., Wang, J., Zheng, Z., Xu, X., Zhai, T., Chen, X., Lu, X., Fu, W., Li, X., Zeng, L., Hu, M., Zhang, Y., and Fan, S.: Variation and trend of nitrate radical reactivity towards volatile organic compounds in Beijing, China, Atmos. Chem. Phys., 23, 8211-8223, https://doi.org/10.5194/acp-23-8211-2023, 2023.

Li, J., Zhai, T., Chen, X., Wang, H., Xie, S., Chen, S., Li, C., Gong, Y., Dong, H., and Lu, K.: Direct measurement of N2O5 heterogeneous uptake coefficients on atmospheric aerosols in southwestern China and evaluation of current parameterizations, Atmos. Chem. Phys., 25, 6395-6406, https://doi.org/10.5194/acp-25-6395-2025, 2025.

Thaler, R. D., Mielke, L. H., and Osthoff, H. D.: Quantification of Nitryl Chloride at Part Per Trillion Mixing Ratios by Thermal Dissociation Cavity Ring-Down

Spectroscopy, Analytical Chemistry. 83, 2761-2766, https://doi.org/10.1021/ac200055z, 2011.

Wang, H., Peng, C., Wang, X., Lou, S., Lu, K., Gan, G., Jia, X., Chen, X., Chen, J., Wang, H., Fan, S., Wang, X., and Tang, M.: $N_2O_5$ uptake onto saline mineral dust: a potential missing source of tropospheric $ClNO_2$ in inland China, Atmos. Chem. Phys., 22, 1845-1859, https://doi.org/10.5194/acp-22-1845-2022, 2022a.

Wang, H. C., Peng, C., Wang, X., Lou, S. R., Lu, K. D., Gan, G. C., Jia, X. H., Chen, X. R., Chen, J., Wang, H. L., Fan, S. J., Wang, X. M., and Tang, M. J.: $N_2O_5$ uptake onto saline mineral dust: a potential missing source of tropospheric $ClNO_2$ in inland China, Atmospheric Chemistry and Physics. 22, 1845-1859, https://doi.org/10.5194/acp-22-1845-2022, 2022b.

Wang, H. C., Yuan, B., Zheng, E., Zhang, X. X., Wang, J., Lu, K. D., Ye, C. S., Yang, L., Huang, S., Hu, W. W., Yang, S. X., Peng, Y. W., Qi, J. P., Wang, S. H., He, X. J., Chen, Y. B., Li, T. G., Wang, W. J., Huangfu, Y. B., Li, X. B., Cai, M. F., Wang, X. M., and Shao, M.: Formation and impacts of nitryl chloride in Pearl River Delta, Atmospheric Chemistry and Physics. 22, 14837-14858, https://doi.org/10.5194/acp-22-14837-2022, 2022c.

Zhang, Y. N., Wang, H. L., Huang, L. B., Qiao, L. P., Zhou, M., Mu, J. S., Wu, C., Zhu, Y. J., Shen, H. Q., Huang, C., Wang, G. H., Wang, T., Wang, W. X., and Xue, L. K.: Double-Edged Role of VOCs Reduction in Nitrate Formation: Insights from Observations during the China International Import Expo 2018, Environmental Science & Technology. 57, 15979-15989, https://doi.org/10.1021/acs.est.3c04629, 2023.